# Adaptive Querying with AI Persona Priors

**Kaizheng Wang** [* 1 2]   **Yuhang Wu** [* 3]   **Assaf Zeevi** [* 3]

## Abstract

We study adaptive querying for learning user-dependent quantities of interest, such as responses to held-out items and psychometric indicators, within tight question budgets. Classical Bayesian design and computerized adaptive testing typically rely on restrictive parametric assumptions or expensive posterior approximations, limiting their use in heterogeneous, high-dimensional, and cold-start settings. We introduce a persona-induced latent variable model that represents a user's state through membership in a finite dictionary of AI personas, each offering response distributions produced by a large language model. This yields expressive priors with closed-form posterior updates and efficient finite-mixture predictions, enabling scalable Bayesian design for sequential item selection. Experiments on synthetic data and WorldValuesBench demonstrate that persona-based posteriors deliver accurate probabilistic predictions and an interpretable adaptive elicitation pipeline.

## 1. Introduction

Many interactive systems must learn about users under severe information constraints. Examples range from market research surveys and psychometrics to recommender systems and preference elicitation, where only a small number of questions can be asked before user fatigue, privacy concerns, or cost constraints intervene. In these settings, the goal is not merely to predict individual responses, but to form calibrated probabilistic beliefs about user-dependent quantities of interest—such as held-out responses, psychometric indicators, or downstream decisions—within tight query budgets. Bayesian adaptive querying provides a natural framework for this problem by explicitly modeling

uncertainty and selecting questions to maximally reduce it.

Despite its conceptual appeal, existing Bayesian design and adaptive testing methods face a practical tension between *expressiveness* and *tractability*. Classical computerized adaptive testing (CAT) and item response theory (IRT) typically rely on low-dimensional parametric latent traits, which can be restrictive when response patterns are heterogeneous and high-dimensional, as in modern recommender systems. Conversely, more flexible Bayesian models often require costly posterior approximations (e.g., nested Monte Carlo or variational inference), which can be difficult to deploy in real-time interactive settings. These challenges are amplified in *cold-start* regimes (Schein et al., 2002), where either the user is new (little or no history) and/or items/questions are new (limited calibration data), precisely when strong and structured priors are most valuable.

Recent advances in large language models (LLMs) suggest a new ingredient: LLMs can simulate plausible human responses when conditioned on rich textual profiles or personas, reproducing response patterns of specific demographic and attitudinal subgroups (Aher et al., 2023; Argyle et al., 2023; Horton, 2023). This capability has spurred interest in using personas for elicitation and adaptive questioning, but most existing approaches treat personas and LLM outputs as heuristic tools rather than as components of a coherent Bayesian model with principled posterior updates and decision-theoretic query selection. This motivates our central question:

*Can we use AI personas to define a simple yet expressive Bayesian prior that supports efficient adaptive querying?*

**Overview.** We propose *Adaptive Querying with AI Persona Priors*. The key idea is to represent user heterogeneity through membership in a finite dictionary of AI personas, where each persona induces a distribution over responses to each question. We obtain these persona–question response distributions *offline* by prompting an LLM; online, for a new user, we initialize a prior over persona membership and update it sequentially as answers arrive. The resulting posterior is a finite mixture with closed-form updates and predictions, enabling Bayesian experimental design (BED) methods and adaptive querying policies to be implemented efficiently. We illustrate the workflow in Figure 1.

---

[*]Equal contribution [1]Department of Industrial Engineering and Operations Research, Columbia University [2]Data Science Institute, Columbia University [3]Decision, Risk, and Operations Division, Columbia Business School. Correspondence to: Kaizheng Wang <kaizheng.wang@columbia.edu>.

*Proceedings of the 43$^{rd}$ International Conference on Machine Learning*, Seoul, South Korea. PMLR 306, 2026. Copyright 2026 by the author(s).

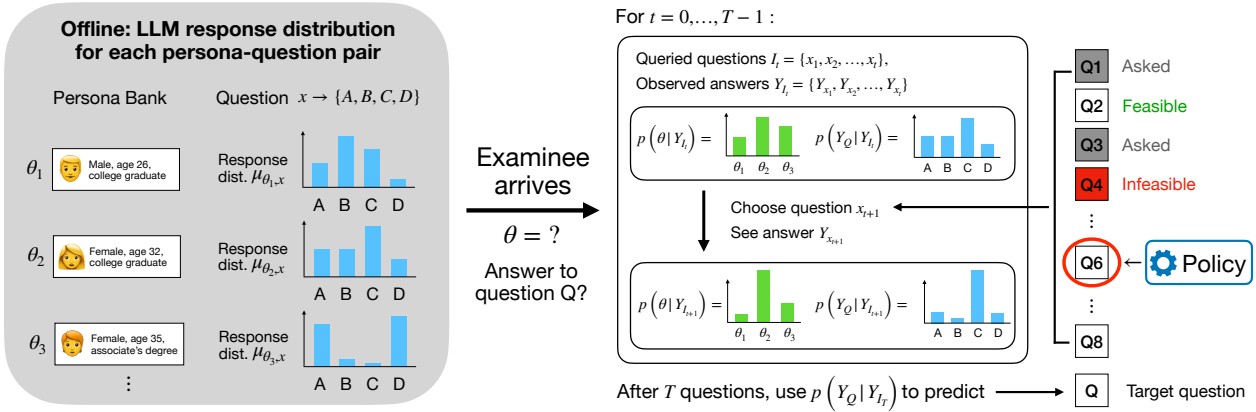

*Figure 1.* Workflow of our persona-based Bayesian adaptive querying. Offline, we collect persona–question response distributions from an LLM for a dictionary of personas. Online, a new user is modeled via a prior over persona membership, which is updated through Bayesian adaptive querying to form posterior beliefs and predictions. After exhausting a budget of questions, we make a probabilistic prediction on the user's answer to a target question.

**Contributions.** Our primary contribution is an *end-to-end recipe* for turning LLM persona outputs into a Bayesian prior that supports closed-form posterior updates and efficient adaptive querying in noisy cold-start settings. Unlike classical CAT/IRT, our persona dictionary and response distributions are obtained entirely offline from an LLM, eliminating the need for task-specific item calibration and making the method immediately deployable. When training users are available, the prior over personas can also be adapted via empirical Bayes. Unlike recent neural BED approaches (Foster et al., 2021; Ivanova et al., 2021), our model retains exact posterior inference, avoiding learned surrogates or policy networks. Concretely, our contributions include:

- **A principled persona prior.** We introduce a *persona-induced latent variable model* in which a user is represented by a member of a finite persona dictionary, with persona–question likelihoods provided by an LLM. This yields a simple but expressive prior over high-dimensional categorical response vectors.

- **Tractable Bayesian inference at scale.** Under categorical questions, the model admits *closed-form posterior updates* over persona membership and *finite-mixture posterior predictions*, avoiding nested Monte Carlo or variational approximations that commonly bottleneck BED in high dimensions.

- **Instantiation of classical adaptive methods.** We demonstrate how standard non-adaptive and adaptive Bayesian design strategies can be implemented efficiently within our persona-based Bayesian model, and note how RL-style formulations fit naturally into the same framework.

- **Empirical study with CAT as a reference point.** On synthetic data and WorldValuesBench, we evaluate persona-based posteriors as probabilistic predictors and compare against classical CAT/IRT baselines. The comparisons illustrate when persona priors can be especially effective, including cold-start user/item regimes that are challenging for calibration-heavy models.

The remainder of the paper is organized as follows. Section 2 formalizes the adaptive querying problem, Section 3 introduces our persona-induced latent variable model and inference methods, and Section 4 presents experiments on synthetic and real data. We conclude with related work and limitations in Section 5, and defer a full literature review to Appendix B.

## 2. Problem Formulation

We study the problem of sequentially querying a user in order to learn about an unknown quantity of interest under a limited budget of questions. Let $Y \in \mathcal{Y}^m$ denote a random vector representing the user's responses to a fixed bank of $m$ questions, where $\mathcal{Y}$ is a response space. User responses are assumed to be intrinsically noisy, and we model $Y$ as a random vector drawn from a prior distribution $p_Y$. Our objective is to infer a target quantity

$$Z \triangleq g(Y),$$

where $g$ is a known mapping. The quantity $Z$ may represent, for example, the user's responses to a subset of unasked questions, a latent categorical label, a real-valued score, or a Bayes-optimal decision derived from $Y$. In many applications, not all questions can be asked due to cost, sensi-

tivity, or operational constraints. We model this by letting $\mathcal{I}_{\text{feas}} \subseteq [m]$ denote the set of feasible question indices.

## 2.1. Setup

**Uncertainty-based objective.** When the prior distribution $p_Y$ is known, it induces a well-defined distribution over the target quantity $Z$. In this case, learning about $Z$ can be formalized as reducing uncertainty in its posterior distribution. Let $U(\cdot)$ denote a real-valued functional that measures uncertainty, such as Shannon entropy, variance, or Gini impurity. With slight abuse of notation, we write $U(X)$ to denote the uncertainty of a random variable $X$ through its distribution. The goal of adaptive querying is to design a policy that minimizes the uncertainty of $Z$ after a limited number of queries.

**Bayesian adaptive querying.** At each time step $t = 1, \ldots, T$ with $T \leq m$, let

$$h_t \triangleq (x_1, Y_{x_1}, \ldots, x_t, Y_{x_t})$$

denote the interaction history, where $x_i \in \mathcal{I}_{\text{feas}}$ is the selected question and $Y_{x_i}$ is the corresponding observed response. We assume that each question can be asked at most once, and denote the set of queried questions by $I_t = \{x_1, \ldots, x_t\}$. Define $h_0 = I_0 = \varnothing$. Conditioning on the history $h_t$, let

$$P_t \triangleq p(Z \mid h_t)$$

denote the posterior distribution of the target quantity. A Bayesian adaptive querying policy $\pi$ selects the next question $x_{t+1} \in \mathcal{I}_{\text{feas}} \setminus I_t$ based on $h_t$, observes the response $Y_{x_{t+1}}$, and continues this process until the budget is exhausted. The performance of $\pi$ is measured by the uncertainty of the final posterior $P_T$. Formally, we seek to design a policy $\pi$ that minimizes the expected posterior uncertainty:

$$\min_{\pi} \; \mathbb{E}[U(P_T)], \tag{1}$$

where the expectation is taken with respect to the randomness in the user's responses, induced jointly by the prior $p_Y$ and the policy $\pi$. We may also write the objective as $E[U(Z|h_T)]$.

## 2.2. Evaluation and Scoring Rules

In practice, the assumed prior $p_Y$ is rarely exact, and posterior beliefs $P_t$ may be misspecified relative to the true data-generating process, making it essential to evaluate probabilistic predictions using statistically principled criteria.

We adopt the framework of *proper scoring rules*. A scoring rule is a function $S(p, z)$ that assigns a numerical score to a predictive distribution $p$ when outcome $z$ is realized. It

is *strictly proper* if the expected score is uniquely maximized when $p$ coincides with the true distribution. Proper scoring rules therefore incentivize calibrated and honest probabilistic predictions.

A classical result establishes a close duality between uncertainty measures and proper scoring rules (McCarthy, 1956; Savage, 1971). In particular, every strictly concave uncertainty functional $U$ induces a strictly proper scoring rule $S$, and conversely, any strictly proper scoring rule $S$ defines an uncertainty functional via its expected negative self-score,

$$U_S(p) \triangleq -\mathbb{E}_{Z \sim p}[S(p, Z)].$$

Canonical examples include Shannon entropy paired with logarithmic scoring and Gini impurity paired with the Brier score (Gneiting & Raftery, 2007). This correspondence ensures that uncertainty-based query selection objectives align naturally with principled evaluation metrics, even under model misspecification.

## 2.3. Approximate Solution Methods

The optimization problem in (1) is combinatorial and generally NP-hard. Consequently, practical solutions rely on approximation methods (Rainforth et al., 2024). We outline several common approaches below; their efficient persona-based instantiations are discussed in Section 3.

**Non-adaptive optimal design.** Classical Bayesian experimental design considers a non-adaptive setting in which all $T$ questions are selected upfront:

$$\min_{I \subseteq \mathcal{I}_{\text{feas}}, |I| = T} \; \mathbb{E}[U(Z \mid Y_I)]. \tag{2}$$

This formulation avoids interaction-dependent computation and can be easier to deploy in practice. However, it ignores user-specific responses observed during querying and is therefore generally less sample-efficient. In practice, greedy forward selection heuristics—analogous to forward feature selection in regression—are often used to approximate (2).

**Greedy adaptive querying.** A widely used adaptive strategy is greedy one-step lookahead. At time $t$, for each candidate question $x \in \mathcal{I}_{\text{feas}} \setminus I_t$, one computes the expected posterior uncertainty after observing its response,

$$\Delta_U(x \mid h_t) \triangleq \mathbb{E}_{Y_x \sim p(\cdot \mid h_t)} \Big[ U(Z \mid h_t, Y_x) \Big], \tag{3}$$

and selects the question that minimizes this quantity. This procedure prioritizes questions with the largest expected immediate reduction in uncertainty. Extensions to multi-step lookahead or tree search are possible but are typically more computationally demanding.

**Reinforcement learning.** The scoring-rule perspective naturally yields a non-myopic reinforcement learning (RL) formulation of adaptive querying. The interaction between the agent and the user defines a finite-horizon episodic decision process, where actions correspond to question selections and observations correspond to responses. We can define the reward in step $t$ as $U(P_{t-1}) - U(P_t)$, which measures the uncertainty reduction. The cumulative reward over $T$ steps is $U(P_0) - U(P_T)$, making the RL objective equivalent to minimizing final posterior uncertainty.

Beyond these approaches, the formulation in (1) also encompasses Thompson sampling-style policies, Bayesian optimization acquisition functions, and other information-theoretic strategies. We leave a systematic comparison of these methods to future work.

## 3. Methodology

The adaptive querying strategies described in Section 2 are agnostic to the choice of prior distribution $p_Y$. In practice, however, their successful deployment hinges on the ability to efficiently compute posterior distributions and predictive likelihoods at each step of the interaction. For general high-dimensional priors with complex dependencies across questions, posterior inference can be intractable, rendering even greedy adaptive methods computationally prohibitive.

This motivates the use of structured probabilistic models that balance *expressiveness*—the ability to capture rich and heterogeneous user response patterns—with *tractability*—the ability to support fast posterior updates and prediction. In this section, we introduce a latent variable model based on AI personas that achieves this balance. The resulting model admits closed-form inference while leveraging LLMs to encode complex prior information.

### 3.1. Persona-Induced Latent Variable Model

A standard way to impose structure on $p_Y$ is through a latent variable $\theta \in \Theta$ that captures user-specific characteristics. We assume conditional independence of responses across questions given $\theta$, yielding the joint model

$$p(\theta, Y) \;=\; p_\theta(\theta) \prod_{i=1}^{m} p(Y_i \mid \theta). \qquad (4)$$

This is a simplifying assumption shared with classical IRT and CAT models that enables closed-form posterior updates. Given observations $Y_{I_t}$, Bayes' rule gives the posterior

$$p(\theta \mid Y_{I_t}) \;\propto\; p_\theta(\theta) \prod_{i \in I_t} p(Y_i \mid \theta), \qquad (5)$$

and the posterior predictive distribution for an unasked question $x$ is

$$p(Y_x \mid Y_{I_t}) \;=\; \int p(Y_x \mid \theta)\, p(\theta \mid Y_{I_t})\, d\theta. \qquad (6)$$

This latent-variable formulation allows posterior predictive sampling via a two-step procedure: first sample $\theta$ from $p(\theta \mid Y_{I_t})$, then sample $Y_x$ from $p(\cdot \mid \theta)$. However, in the context of Bayesian adaptive querying, the predictive integral in (6) is typically nested inside expectations over future observations (cf. (1)), leading to repeated high-dimensional integrations at every decision step.

This challenge is well known in Bayesian experimental design and has motivated approaches such as nested Monte Carlo estimation (Rainforth et al., 2018) and variational approximations (Foster et al., 2019). While effective in some settings, these methods remain computationally intensive and introduce accuracy-efficiency trade-offs. This motivates the search for a latent variable model that is both expressive and admits efficient posterior updates.

Recent advances in LLMs provide a compelling answer. LLMs can generate coherent, human-like responses when conditioned on descriptive profiles or personas, suggesting a natural way to encode rich prior beliefs about user behavior. Suppose we are given a dictionary of $n$ AI personas with profiles $\xi_1, \ldots, \xi_n$. For each persona and question, we can query an LLM conditioned on the persona profile to obtain an estimated response distribution.

If the persona dictionary is sufficiently representative, it is reasonable to model a new user as one (or a mixture) of these personas. Accordingly, we define the latent variable as persona membership,

$$\theta \in \{1, 2, \ldots, n\},$$

and interpret the user as being drawn from persona $\theta$.[1] We posit a prior $p(\theta)$ and define the item-response model as

$$Y_x \mid \theta \;=\; \mathsf{LLM}(\xi_\theta, x).$$

The response distribution $\mathsf{LLM}(\xi_\theta, x)$ can be obtained in various ways, including prompting, log-probability extraction, or calibrated sampling; see Appendix C for details. This construction turns LLM-based personas into an explicit probabilistic prior rather than a heuristic simulation tool, enabling principled Bayesian inference.

**Categorical questions.** For clarity and concreteness, we focus on the setting where all questions have $K$ categorical responses, so that $Y \in \{1, 2, \ldots, K\}^m$. For each persona-

---

[1]While the prior over $\theta$ corresponds to a hard assignment, the posterior is soft and supports mixture-like inference.

question pair, we model the response distribution as categorical with parameter

$$\mu_{\theta,x} = (\mu_{\theta,x,1}, \ldots, \mu_{\theta,x,K}) \in \Delta^{K-1},$$

so that

$$Y_x \mid \theta = \mathsf{LLM}(\xi_\theta, x) = \mathrm{Categorical}(\mu_{\theta,x}).$$

Under this model, Bayesian inference admits closed-form expressions. The posterior over persona membership after observing $Y_{I_t}$ is

$$p(\theta \mid Y_{I_t}) \propto p(\theta) \prod_{i \in I_t} \mu_{\theta,i,Y_i}, \qquad (7)$$

which can be normalized efficiently since $\theta$ ranges over a finite set. The posterior predictive distribution for an unasked question $x$ is then

$$p(Y_x = k \mid Y_{I_t}) = \sum_{\theta=1}^{n} \mu_{\theta,x,k}\, p(\theta \mid Y_{I_t}). \qquad (8)$$

**Discussion.** This finite-mixture structure combines the expressiveness of LLM-generated response distributions with the computational simplicity of discrete latent variable models. Structurally, the persona-induced model is a finite mixture model, connecting it to latent class analysis (Goodman, 1974). However, it differs in that the mixture components are not estimated from task-specific data but are instead defined by LLM-elicited response distributions. Moreover, the latent variable $\theta$ has a clear semantic interpretation as persona membership, enabling downstream tasks such as user clustering, response simulation, and group-level analysis. Importantly, the framework is model-agnostic and can be instantiated with *any* pre-trained or fine-tuned LLM.

### 3.2. Non-Adaptive Optimal Design

We first consider non-adaptive Bayesian optimal design under the persona-induced model. In this setting, all $T$ questions are selected *a priori* before observing any responses, corresponding to the batch formulation of Bayesian experimental design. For a candidate set $I$, the expected posterior uncertainty can be written as

$$\mathbb{E}[U(Z \mid Y_I)] = \sum_{y_I \in \mathcal{Y}^{|I|}} p(Y_I = y_I)\, U(Z \mid Y_I = y_I), \quad (9)$$

where the marginal likelihood is

$$p(Y_I = y_I) = \sum_{\theta=1}^{n} p(\theta) \prod_{i \in I} \mu_{\theta,i,y_i}.$$

Although (9) is available in closed form, selecting the optimal subset in (2) remains a combinatorial optimization

problem and is generally NP-hard. A common approximation is greedy forward selection: starting from $I_0 = \varnothing$, at each step select

$$x_{t+1} \in \underset{x \in \mathcal{I}_{\mathrm{feas}} \setminus I_t}{\mathrm{argmin}}\; \mathbb{E}\big[U\big(Z \mid Y_{I_t \cup \{x\}}\big)\big]. \qquad (10)$$

Unlike adaptive querying, this expectation is computed before any responses are observed, and the resulting question set is fixed across users.

---

**Algorithm 1** Greedy Non-Adaptive Optimal Design

---

**Require:** Budget $T$; feasible questions $\mathcal{I}_{\mathrm{feas}}$; prior $p(\theta)$; likelihoods $\{\mu_{\theta,x}\}$; uncertainty functional $U(\cdot)$
1: Initialize $I_0 \leftarrow \varnothing$
2: **for** $t = 0, 1, \ldots, T-1$ **do**
3:     **for all** $x \in \mathcal{I}_{\mathrm{feas}} \setminus I_t$ **do**
4:         Compute expected posterior uncertainty

$$\Delta_U^{\mathrm{batch}}(x \mid I_t) = \mathbb{E}\big[U\big(Z \mid Y_{I_t \cup \{x\}}\big)\big]$$

        using (9)
5:     **end for**
6:     Select $x_{t+1} \leftarrow \mathrm{argmin}_{x \in \mathcal{I}_{\mathrm{feas}} \setminus I_t} \Delta_U^{\mathrm{batch}}(x \mid I_t)$
7:     Update $I_{t+1} \leftarrow I_t \cup \{x_{t+1}\}$
8: **end for**
9: **Return:** Fixed question set $I_T$

---

Algorithm 1 summarizes this procedure. Non-adaptive designs are simple to deploy and avoid online interactive computation. However, they cannot tailor queries to individual users and are thus usually less sample-efficient than adaptive methods. At the same time, they may be more robust to model misspecification and overly aggressive adaptivity.

### 3.3. Greedy Adaptive Querying

Under the categorical persona model, greedy adaptive querying from Section 2 becomes particularly efficient. The one-step lookahead objective in (3) reduces to

$$\Delta_U(x \mid h_t) = \sum_{k=1}^{K} p(Y_x = k \mid Y_{I_t})\, U\big(Z \mid h_t, Y_x = k\big), \tag{11}$$

where each term is computed using (7) and (8). Algorithm 2 summarizes the resulting procedure.

**Connection to collaborative filtering (CF).** Our approach bears a resemblance to CF and lookalike modeling, which also leverage population-level patterns to predict individual preferences (Su & Khoshgoftaar, 2009). However, they differ in several important respects. First, our model is a *generative Bayesian model* with an explicit latent variable and closed-form posterior updates, rather than a matrix-factorization approach. Second, our persona prior requires

**Algorithm 2** Greedy Adaptive Querying

---

**Require:** Budget $T$; feasible questions $\mathcal{I}_{\text{feas}}$; prior $p(\theta)$;
likelihoods $\{\mu_{\theta,x}\}$; uncertainty functional $U(\cdot)$

1: $I_0 \leftarrow \varnothing$, $Y_{I_0} \leftarrow \varnothing$
2: **for** $t = 0, 1, \ldots, T - 1$ **do**
3:    **for all** $x \in \mathcal{I}_{\text{feas}} \setminus I_t$ **do**
4:       Compute $p(Y_x \mid Y_{I_t})$ using (7) and (8)
5:       Compute $\Delta_U(x \mid Y_{I_t})$ using (11)
6:    **end for**
7:    Select $x_{t+1} \leftarrow \operatorname{argmin}_{x \in \mathcal{I}_{\text{feas}} \setminus I_t} \Delta_U(x \mid Y_{I_t})$
8:    Query question $x_{t+1}$ and observe answer $Y_{x_{t+1}}$
9:    Update $I_{t+1} \leftarrow I_t \cup \{x_{t+1}\}$, $Y_{I_{t+1}} \leftarrow (Y_{I_t}, Y_{x_{t+1}})$
10: **end for**
11: **Return:** Observed answers $Y_{I_T}$

---

*no historical response data* from the target population. It is constructed entirely from LLM-generated persona profiles, making it suitable for cold-start settings where CF methods have insufficient data. Third, our framework supports *decision-theoretic query selection*, actively choosing which questions to ask to reduce posterior uncertainty, rather than passively processing available ratings.

## 4. Experiments

We evaluate the proposed persona-based Bayesian adaptive querying framework on both synthetic and real users from WorldValuesBench, with classical computerized adaptive testing (CAT) methods as reference baselines.[2]

### 4.1. Datasets and Persona Construction

**WorldValuesBench (Zhao et al., 2024).** WorldValues-Bench contains survey responses from over 94,000 participants to 290 questions on values and beliefs (e.g., family, politics, religion, work, and society). We restrict attention to ordinal Likert-style questions with four categories and filter out respondents with more than 20% missing answers. The resulting dataset contains 91 questions and 88,459 users, with an overall missing rate of 2.6%. For a given user, if the response to a question is missing, we treat that question as infeasible for that user (i.e., it cannot be queried). During evaluation, metrics are computed only on user–question pairs with observed ground-truth responses. This convention ensures that users with more missing data naturally retain higher posterior uncertainty, since fewer observations are available to update their latent membership.

**Persona dictionary and response distributions.** We use the Twin-2K-500 persona bank (Toubia et al., 2025) as a

---

dictionary of $n = 2{,}058$ latent profiles. Each persona corresponds to a real U.S. participant whose responses to over 500 questions spanning demographic, psychological, economic, and behavioral domains have been collected. The dictionary has three desirable properties: (i) *diversity*—personas cover a broad range of demographic backgrounds and attitudinal profiles; (ii) *domain coverage*—the underlying question bank spans topics well beyond WorldValuesBench, providing rich conditioning information; and (iii) *grounding*—each persona is anchored to a real individual's response pattern, reducing the risk of generating unrealistic or incoherent profiles. These personas do not correspond to real users in WorldValuesBench; instead, they provide a structured, interpretable prior over response patterns. For each persona $\xi_\theta$ and each question $x$, we prompt an LLM (GPT-5-mini) to produce a categorical distribution over the four Likert responses, yielding parameters $\mu_{\theta,x} \in \Delta^3$. Appendix C details prompting, parsing, and quality control.

**Synthetic users (well-specified prior).** To study behavior under correct specification, we generate synthetic users from the persona model. For each synthetic user $j$, we sample a single persona $\theta^{(j)} \sim p(\theta)$ and then sample responses as $Y_x^{(j)} \sim \text{Categorical}(\mu_{\theta^{(j)},x})$ for each question $x$.

### 4.2. Experimental Protocol and Evaluation

We split users into training (80%) and test (20%) sets. All evaluations are performed on held-out test users. For each test user, the algorithm sequentially selects questions from a feasible set and observes the corresponding ground-truth responses; after a budget of $T$ queries, we evaluate the resulting posterior predictive distribution on target questions.

**Target questions and feasible set.** We consider a held-out prediction task in which a small subset of questions $I^\star$ are designated as *targets*, i.e., $g(Y) = Y_{I^\star}$, while the remaining questions constitute the feasible set $\mathcal{I}_{\text{feas}}$. This setting captures applications where a small set of key indicators is of primary interest and must be inferred from a limited interactive budget. In our experiments, we randomly select (the same) 5 questions as targets for each test user, leaving the remaining 86 questions as the feasible set.

**Metrics.** Let $\hat{p}_{u,q}$ denote the predictive distribution for user $u$ on target question $q \in I^\star$, and let $y_{u,q}$ denote the realized response. We report the following metrics averaged over all $(u, q)$ pairs in the test set:

- **Log loss:** $-\log \hat{p}_{u,q}(y_{u,q})$.

- **Brier score:** $\sum_{k=1}^{K} (\hat{p}_{u,q}(k) - \mathbf{1}\{y_{u,q} = k\})^2$.

- **Ordinal MSE:** squared error between the posterior

mean under $\hat{p}_{u,q}$ and the ordinal-coded outcome (categories mapped to $\{0, 1, 2, 3\}$).

## 4.3. Methods Compared

**Persona-based querying policies.** Our persona-based methods use the sum of Shannon entropies of the target marginals, $U(P_t) = \sum_{x' \in I^\star} H(Y_{x'} \mid h_t)$, as the uncertainty functional.[3] In the held-out task, the one-step lookahead objective becomes

$$\Delta_U(x \mid h_t) = \sum_{k=1}^{K} \Big[ p(Y_x = k \mid Y_{I_t}) \\ \cdot \sum_{x' \in I^\star} H(Y_{x'} \mid h_t, Y_x = k) \Big],$$

i.e., the expected posterior sum of marginal target entropies after querying $x$.

**Prior specification.** For experiments on real users, we learn the prior $p(\theta)$ from training users via empirical Bayes by maximizing the marginal likelihood

$$\max_{p(\theta) \in \Delta^{n-1}} \sum_{j=1}^{N} \log \left( \sum_{\theta} p(\theta) \, p(Y^{(j)} \mid \theta) \right).$$

We optimize this objective with an EM algorithm: the E-step computes responsibilities $\gamma_{j,\theta} \propto p(\theta) \, p(Y^{(j)} \mid \theta)$, and the M-step updates $p(\theta) = \frac{1}{N} \sum_{j=1}^{N} \gamma_{j,\theta}$. The learned prior downweights personas that are rarely matched to real users, concentrating mass on the most relevant region of persona space and mitigating the misspecification inherent in applying a synthetic persona dictionary to a real population. For synthetic users, where data is generated from the persona model, we use a uniform prior.

In addition to Algorithm 1 and Algorithm 2, we compare several additional persona-based baselines. The **Random** strategy adaptively selects feasible questions uniformly at random at each step, while **Random Fixed** selects a fixed set of $T$ questions uniformly at random for all users. We also include a **Full** baseline that queries all feasible questions; this serves as an oracle upper bound on available information, though not necessarily on predictive performance when the persona prior is misspecified.

**CAT baselines.** We implement classical polytomous CAT methods based on item response theory (IRT). Specifically,

we consider the graded response model (GRM) and generalized partial credit model (GPCM), each in both one-dimensional and multidimensional variants (MGRM/MG-PCM) (Samejima, 1969; Muraki, 1992; Wainer et al., 2000; Yao & Schwarz, 2006; Reckase, 2009; Van der Linden & Glas, 2010). For each model, we fit item parameters on the training users via marginal maximum likelihood (EM) and perform inference with a grid-based posterior over latent traits. Since existing open-source CAT libraries do not robustly support multidimensional polytomous settings, we implement these baselines from scratch; details are in Appendix F.7.

The two families of methods differ sharply in what they require from training data. CAT baselines must calibrate item parameters for *every item in the bank*, which requires a large number of user responses to each item. If new items are introduced or the item bank changes, recalibration is necessary. In contrast, persona-based methods obtain item-level response distributions entirely from the LLM—no observed responses to those items are needed. The only component learned from training data is the prior $p(\theta)$ over personas, a single $n$-dimensional weight vector that does not depend on the identity of individual items.

In our experiments, we provide CAT with more than 70,000 training users—sufficient for reliable item calibration—making this a generous test of CAT performance. Persona-based methods use the same training split, but only to fit the persona prior. The goal of this setup is to show that, even under favorable conditions for CAT, persona-based methods remain competitive with or superior to this well-established and effective approach. When calibration data is scarce or unavailable—as in cold-start item regimes where new questions must be deployed without prior response data—CAT simply cannot be applied, whereas persona-based methods can incorporate new items immediately via LLM prompting.

## 4.4. Results

### 4.4.1. SYNTHETIC USERS (WELL-SPECIFIED MODEL)

We first evaluate on 100,000 synthetic users sampled from the persona model, where the prior is correctly specified. Figure 2a reports log loss versus query budget; numerical values at representative budgets are given in Table 3 (Appendix E.1). As expected, persona-based methods dominate CAT baselines in the well-specified setting: the persona model matches the data-generating process, while IRT-based CAT is structurally misspecified. In the left panel of Figure 2, performance improves approximately monotonically with budget for all persona-based methods, and the **Full** curve serves as an upper bound, suggesting that with synthetic data all questions are informative about the target questions. Greedy achieves the fastest reduction in log loss, confirming it as a strong and simple adaptive heuristic. Ad-

---

[3]Thus, for the vector-valued target $Z = Y_{I^\star}$ in the held-out task, we evaluate uncertainty additively across target coordinates rather than through the joint entropy of $Z$. The joint entropy is summed over exponentially many terms in the number of target questions, while the sum of marginal entropies is more scalable and still captures the overall uncertainty in the target.

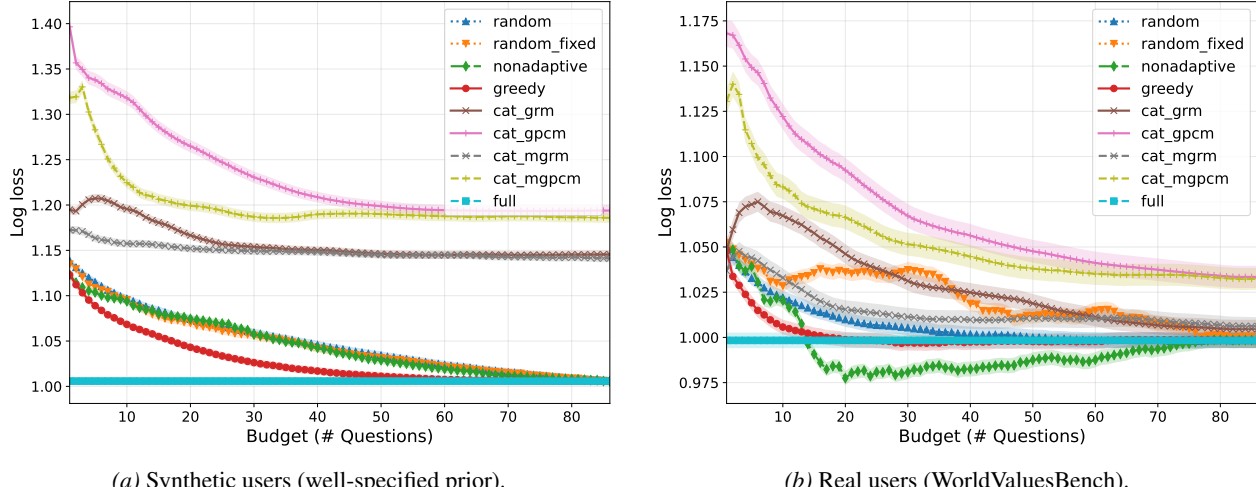

*(a)* Synthetic users (well-specified prior).

*(b)* Real users (WorldValuesBench).

*Figure 2.* Log loss versus query budget. Curves denote mean log loss averaged over all test user–target-question pairs; shaded regions indicate 95% confidence intervals. Left: when the persona prior is correctly specified, persona-based methods substantially outperform CAT baselines, with greedy achieving the fastest error reduction. Right: under model misspecification on real data, persona-based methods still outperform CAT; greedy performs best at small budgets, while non-adaptive designs can overtake at larger budgets. Numerical values at representative budgets are reported in Tables 3 and 4 (Appendix E.1).

ditional metrics (Brier score and ordinal MSE) show the same qualitative behavior (Appendix E.1).

### 4.4.2. REAL USERS (MISSPECIFIED MODEL)

We next evaluate on held-out WorldValuesBench users (Figure 2b; values at representative budgets in Table 4, Appendix E.1). Compared to synthetic users, gains over CAT persist but are smaller, consistent with inevitable model misspecification on real data; Brier score and ordinal MSE show the same pattern (Appendix E.1).

**Efficiency-robustness tradeoff.** A closer look at Figure 2b reveals two notable patterns. First, the **Full** curve does not always yield the best predictive performance, and the non-adaptive curve can degrade as $T$ increases beyond a moderate budget. Second, greedy performs best at small budgets (e.g., $T \leq 15$ in our experiments) but is overtaken by the non-adaptive design at larger ones. Both patterns reflect *model misspecification*: additional queried questions can sometimes be weakly informative or even misleading for predicting the target set. The crossover between greedy and non-adaptive designs further reveals an *efficiency-robustness tradeoff*. Under correct specification, greedy one-step lookahead is near-optimal for entropy-based uncertainty-reduction objectives; under misspecification, however, greedy can overcommit: by selecting questions that are maximally informative under the current (misspecified) posterior, it may narrow the posterior prematurely onto an incorrect region of persona space. Subsequent questions are then chosen to refine an already-biased posterior, producing a cascade of locally optimal but globally subop-

timal selections. In contrast, non-adaptive designs select a diverse, user-independent question set that hedges against misspecification by not conditioning on potentially misleading intermediate observations.

### 4.5. Ablation Studies

We probe the robustness of persona priors along three axes that target the most plausible failure modes: dictionary granularity (does using fewer personas hurt?), distributional richness (does the full LLM-elicited distribution carry information beyond its mode?), and calibration (are the LLM-elicited probabilities already well-tuned for the adaptive querying objective?). Full results, including detailed protocols and tables for the **nonadaptive** and **greedy** methods (Tables 9 and 10), are reported in Appendix E.2. (i) **Dictionary clustering.** Compressing the $n = 2,058$ Twin-2K-500 personas into a smaller set of prototypes via prior-weighted $k$-means leaves performance essentially unchanged down to ∼200 clusters and only mildly degraded at 50, indicating that moderate compression does not hurt predictive accuracy. (ii) **Distributional shape vs. deterministic-plus-noise.** Replacing the LLM-elicited distributions with a single mode response plus uniform noise of mass $\varepsilon \in \{0.1, 0.3\}$ degrades performance substantially across all methods and budgets. This confirms that the full distributional shape of the LLM-elicited responses carries information beyond the modal answer. (iii) **Temperature scaling.** Both sharpening ($\tau = 0.5$) and softening ($\tau = 2$) of the LLM-elicited distributions degrade performance, suggesting the original distributions are already roughly well-calibrated for the adaptive querying objective.

## 4.6. Runtime Comparison

Table 1 reports wall-clock runtimes in minutes for all methods on the real WorldValuesBench dataset. All implementations are optimized with standard techniques including contiguous NumPy arrays, Numba JIT compilation, and Joblib parallelization, and all timings were measured on a single Apple MacBook Pro (M1 chip, 8-core CPU/GPU, 16 GB memory). The table separates *inference* (online computation on test users) from *fitting* (offline model calibration on training users). All persona-based methods share a single empirical Bayes prior fitting step, described in the prior-specification paragraph above, which runs an EM algorithm with 100 iterations, completing in 3.98 minutes. Implementation details for CAT methods are in Appendix F.7.

*Table 1.* Runtime comparison on real WorldValuesBench ($n_{\text{train}} = 70{,}767$, $n_{\text{test}} = 17{,}692$; $T = 86$); reported times are in minutes. Inference = online computation on test users; Fitting = offline calibration on training users. [†]Persona-based methods share a single fitting step; the Total column reports the cost of running each method individually.

| Method | Inference | Fitting | Total |
|---|---|---|---|
| full | 0.46 | 3.98[†] | 4.44 |
| random_fixed | 0.47 | 3.98[†] | 4.45 |
| nonadaptive | 0.50 | 3.98[†] | 4.48 |
| random | 0.50 | 3.98[†] | 4.48 |
| greedy | 40.36 | 3.98[†] | 44.34 |
| CAT-GRM | 10.05 | 10.64 | 20.69 |
| CAT-GPCM | 7.85 | 14.33 | 22.18 |
| CAT-MGRM ($D=3$) | 27.52 | 67.82 | 95.34 |
| CAT-MGPCM ($D=3$) | 34.61 | 124.42 | 159.03 |

Non-adaptive persona-based methods complete inference for all 17,692 test users in under one minute, totalling under five minutes once the shared 3.98-minute prior-fitting step is included; **greedy** adds ∼40 minutes of inference, since it recomputes the one-step lookahead at every step, but its ∼44-minute total remains practical at this scale, and the prior-fitting cost is amortized across all persona strategies. CAT baselines, by contrast, must be calibrated separately: unidimensional GRM/GPCM take ∼20–22 minutes each, while the multidimensional MGRM and MGPCM total ∼95 and ∼159 minutes, dominated by EM fitting on a $D = 3$ Cartesian grid—already 4–7× slower than 1D CAT, with cost growing exponentially in $D$. This exposes a fundamental *expressiveness–scalability tradeoff*: although multidimensional CAT yields improved predictions over unidimensional variants (Tables 3 and 4), scaling to higher dimensions is computationally prohibitive, whereas the persona-based model achieves its expressiveness through a large dictionary of $n = 2{,}058$ semantically grounded LLM-powered personas while preserving lightweight closed-form inference.

## 5. Discussion

Our results suggest that AI personas offer a practical middle ground between classical parametric latent variable models and fully black-box generative approaches. By encoding rich prior knowledge through LLM-simulated persona–question response distributions, the resulting model remains expressive while admitting closed-form Bayesian updates and efficient predictive inference. This tractability enables the direct use of standard Bayesian experimental design and adaptive querying methods without resorting to expensive posterior approximations.

**Related work.** Our framework relates to Bayesian experimental design (Lindley, 1956; Rainforth et al., 2024), computerized adaptive testing and item response theory (Wainer et al., 2000; Van der Linden & Glas, 2010; Reckase, 2009), latent class and finite mixture models (Goodman, 1974; McLachlan & Peel, 2000), and recent work on LLMs as human behavior simulators (Argyle et al., 2023; Horton, 2023; Leng et al., 2024) and adaptive natural-language elicitation (Piriyakulkij et al., 2023; Handa et al., 2024; Kobalczyk et al., 2025; Wang et al., 2025). In contrast to neural BED methods that replace exact inference with learned surrogates (Foster et al., 2021; Ivanova et al., 2021), CAT/IRT methods that rely on low-dimensional parametric traits and costly per-item calibration, and LLM-based works that treat persona outputs as black-box simulators or policy networks, we use LLM persona outputs as components of an explicit Bayesian prior with closed-form posterior updates. See Appendix B for a more detailed discussion of the related literature.

**Comparison with CAT and cold-start regimes.** While CAT provides a natural baseline, its reliance on low-dimensional latent traits and offline calibration limits its effectiveness in cold-start user or item settings. In contrast, persona-based priors inject structured prior information that can be leveraged immediately, even when personas are not derived from the evaluation dataset. The observed gains therefore arise not from fitting a more complex model, but from better prior specification within a Bayesian framework.

**Limitations and outlook.** Our approach depends on the quality and diversity of the persona dictionary as well as the fidelity of LLM-generated response distributions. Two future directions are natural. First, learning or refining the persona dictionary online from observed users—e.g., adding personas when posterior mass concentrates near the boundary of the existing set—would address dictionary misspecification more directly. Second, extending the framework to richer response types (continuous, ordinal with covariates, free-text) and to longer-horizon RL-style policies is a promising direction for deployment in practice.

## Acknowledgements

Kaizheng Wang's research is supported by NSF grant DMS-2515679.

## Impact Statement

This paper presents work whose goal is to advance the field of Machine Learning. There are many potential societal consequences of our work, none of which we feel must be specifically highlighted here.

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

# A. Notation

Table 2 summarizes key notation used throughout the paper.

*Table 2.* Summary of notation.

| Symbol | Description |
| --- | --- |
| $m$ | Number of questions in the bank |
| $K$ | Number of response categories per question |
| $\mathcal{Y} = \{1, \ldots, K\}$ | Response space |
| $Y \in \mathcal{Y}^m$ | Random response vector |
| $Y_x$ | Response to question $x$ |
| $\mathcal{I}_{\text{feas}}$ | Feasible question set |
| $I^\star$ | Target question set |
| $T$ | Query budget |
| $h_t$ | Interaction history at step $t$ |
| $P_t = p(Z \mid h_t)$ | Posterior distribution at step $t$ |
| $U(\cdot)$ | Uncertainty functional |
| $S(p, z)$ | Scoring rule |
| $n$ | Number of personas in dictionary |
| $\theta \in \{1, \ldots, n\}$ | Persona membership (latent variable) |
| $\xi_\theta$ | Textual profile of persona $\theta$ |
| $\mu_{\theta,x} \in \Delta^{K-1}$ | Persona–question response distribution |
| $p(\theta)$ | Prior over persona membership |
| $p(\theta \mid Y_{I_t})$ | Posterior over persona membership |
| $\Delta_U(x \mid h_t)$ | Expected posterior uncertainty of querying $x$ |

# B. Extended Related Work

This appendix expands on the brief related-work overview given in Section 5.

**Bayesian experimental design (BED).** BED dates back to the seminal work of Lindley (1956) and has since developed into a rich and mature literature (Chaloner & Verdinelli, 1995; Rainforth et al., 2024). The central idea is to select experiments or queries that optimize an information-theoretic or decision-theoretic objective under a Bayesian model. While conceptually powerful, classical BED methods are often computationally demanding, typically requiring nested Monte Carlo estimation or variational approximations of posterior quantities. As a result, even approximate implementations can be expensive at scale, and exact posterior inference is rarely tractable in high-dimensional settings. Recent work has pursued neural and amortized variants of sequential BED—including mutual-information neural estimation (Kleinegesse & Gutmann, 2020) and learned design policies for real-time deployment (Foster et al., 2021; Ivanova et al., 2021)—but these approaches replace exact inference with learned surrogates or policy networks. In contrast, our persona-induced mixture model retains closed-form posterior updates while leveraging the expressiveness of LLM-generated response distributions. Moreover, the classical BED literature has traditionally relied on parametric statistical models and does not leverage modern generative models as components of the prior or likelihood.

**Active learning and noisy Bayesian querying.** Active learning has a long history, with a comprehensive overview provided by Settles (2009). Our problem is most closely related to Bayesian active learning with noisy observations, where the learner adaptively selects queries to reduce uncertainty about latent structure. Some prior works consider conceptually related problems but differ substantially in formulation or assumptions. For example, Jedynak et al. (2012) study a setting where queries ask whether an item belongs to a proposed set, which does not directly apply to our multi-question, multi-response framework. The EC2 framework of Golovin et al. (2010) considers adaptive querying over a hypothesis space, but is primarily designed for noiseless responses and a relatively small number of hypotheses, making it unsuitable for our setting with a large pool of personas and inherently noisy responses. More broadly, several works in noisy Bayesian active learning, such as Naghshvar et al. (2012), rely on assumptions that no two hypotheses are indistinguishable forever. In contrast, in our setting, different personas may remain probabilistically indistinguishable even after exhausting the querying budget. Our formulation is also related to best-arm identification in bandit problems, but differs in that our objective is not to identify a single optimal arm with i.i.d. rewards, but rather to minimize a general posterior objective functional that may depend on high-dimensional response distributions or downstream decision quality.

**CAT and IRT.** Computerized adaptive testing (CAT) and item response theory (IRT) provide a well-established framework for adaptively selecting test items to efficiently estimate a test-taker's latent trait (Wainer et al., 2000; Van der Linden & Glas, 2010; Lord, 2012). Bayesian approaches to CAT can often be viewed as special cases of BED, with objectives such as posterior variance reduction or information maximization. However, classical CAT and item response theory (IRT) models typically rely on low-dimensional latent variables—often a single scalar ability parameter—and parametric item response functions. These modeling assumptions can be restrictive when user characteristics and response patterns are complex or heterogeneous. Furthermore, posterior updates and predictive likelihoods become intractable in higher-dimensional extensions (Reckase, 2009) or nonparametric variants, limiting scalability. In practice, CAT methods also require a costly offline calibration phase to fit item parameters from large datasets (Bock & Aitkin, 1981), which may not transfer well across domains.

**Latent class analysis and finite mixtures.** Our model is also related to latent class analysis (Goodman, 1974) and finite mixture models (McLachlan & Peel, 2000) for multivariate categorical data. The closest structural analogy is a discrete latent class model with class-conditional question-response probabilities. While classical LCA estimates both the class proportions and the class-conditional distributions entirely from respondent data, our approach specifies the class dictionary and class-conditional distributions *offline* using LLM-generated personas. This distinction is critical for cold-start settings where respondent data for a new item is limited or even unavailable. More broadly, our approach connects to a recent line of work on using generative AI to articulate Bayesian priors. O'Hagan & Ročková (2025) propose taking a generative AI model as the base measure of a Dirichlet process prior on the data-generating distribution, and performing nonparametric loss-based inference via a parallelizable posterior bootstrap. Our setting differs in that LLM persona prompting yields a finite, interpretable mixture prior tailored to multi-question, multi-response elicitation, which admits closed-form posterior updates suitable for adaptive querying.

**LLMs as human behavior simulators.** A growing body of work investigates the use of LLMs to simulate human survey responses and behavioral patterns. Early studies demonstrated that LLMs can replicate aggregate response distributions of demographic subgroups (Argyle et al., 2023; Aher et al., 2023) and reproduce patterns observed in economic experiments (Horton, 2023). Subsequent work has examined which opinions and values are encoded in LLMs (Santurkar et al., 2023; Scherrer et al., 2023), and whether LLMs can serve as reliable proxies for human subjects (Gao et al., 2025; Hullman et al., 2026). These investigations reveal both promise and systematic pitfalls: LLM-generated persona responses can capture meaningful variation across subpopulations, but distortions arise from training data biases and the gap between text generation and genuine human cognition (Li et al., 2025; Peng et al., 2026). Recent efforts have sought to close this gap through fine-tuning on survey data (Cho et al., 2024; Cao et al., 2025), mixture-of-personas architectures for population-level simulation (Leng et al., 2024; Bui et al., 2025; Wang et al., 2026), synthetic control framework for simulation calibration (Fan et al., 2026), and formal frameworks for quantifying the information content of LLM-simulated respondents relative to real humans (Huang et al., 2025; Iyengar et al., 2025). Our approach contributes to this line of work by showing how LLM-generated persona response distributions can serve not merely as simulation outputs but as components of an explicit Bayesian prior that supports principled inference and adaptive decision-making.

**LLMs for adaptive querying.** Complementary to their use as simulators, LLMs have also been explored as adaptive natural-language elicitation systems (Piriyakulkij et al., 2023; Handa et al., 2024; Hu et al., 2024; Mazzaccara et al., 2024; Kobalczyk et al., 2025). These methods typically assume a finite hypothesis set with deterministic or nearly deterministic likelihoods, which in our framework would correspond to a noiseless setting with a small number of personas. In contrast, our setting features inherently stochastic responses where even the true persona's response distribution assigns non-trivial probability to multiple categories, making posterior concentration fundamentally slower and principled uncertainty quantification essential. Wang et al. (2025) propose an adaptive elicitation framework using a meta-learned predictive language model to select questions that maximize simulated future information gain. Our approach differs by maintaining an explicit finite latent persona prior with closed-form Bayesian posterior updates, rather than relying on predictive uncertainty from a neural sequence model.

# C. Obtaining Response Distributions from LLMs

The persona-induced latent variable model requires, for each persona $\xi_\theta$ and question $x$, a probabilistic response model $\text{LLM}(\xi_\theta, x)$, i.e., a distribution over the possible answers to $x$ when conditioned on persona $\xi_\theta$. While LLMs are typically accessed as conditional text generators, there are multiple ways to obtain or approximate such response distributions. We

briefly summarize several common strategies and discuss their trade-offs.

- **Direct distribution elicitation.** One approach is to directly prompt the LLM to output a probability distribution over the admissible responses (e.g., normalized probabilities over Likert categories). This method is simple and inexpensive, and works well when the response space is small and well-defined. However, the resulting distributions may be poorly calibrated or sensitive to prompt phrasing, and there is limited theoretical grounding for treating the reported probabilities as true likelihoods.

- **Logit-based extraction.** When available, one can extract next-token logits corresponding to each admissible response and normalize them to form a distribution. This approach provides a more direct connection to the underlying language model and avoids heuristic prompting. However, access to token-level logits is restricted or unavailable for many state-of-the-art models, and mapping natural-language responses to token probabilities can be nontrivial.

- **Repeated sampling.** Another option is to sample multiple responses from the LLM under a fixed prompt and estimate an empirical distribution over answers. Because persona–question pairs are independent, this procedure can be performed offline and parallelized. Nonetheless, achieving low-variance estimates may require a large number of samples, making this approach computationally expensive at scale.

- **Deterministic response with injected noise.** A simpler alternative is to take a deterministic (e.g., temperature-$0$) response and inject synthetic noise to form a distribution. While computationally cheap, this method often produces unrealistic or overly concentrated distributions, particularly when the response space is multi-modal or when subtle preference uncertainty matters.

In our experiments, we adopt the direct distribution elicitation approach, as it provides a practical trade-off between computational cost and expressiveness for small categorical response spaces. We also perform ablations that employ the deterministic-plus-noise strategy. We emphasize, however, that our framework is agnostic to the specific method used to obtain $\text{LLM}(\xi_\theta, x)$, and any approach that yields a valid conditional distribution can be plugged into the model.

## D. Prompting Details for LLM Response Distributions

To obtain the response distributions $\text{LLM}(\xi_\theta, x)$ for each persona-question pair, we use the following prompt template when querying GPT-5-mini.

---

**System Prompt**

You are an expert in simulating human survey responses. You will be given:

- a detailed persona profile describing a human's values, beliefs, and background;

- a survey question with ordinal response options numbered 1 to 4.

Your task is to predict the persona's \*response distribution\* to the question.

Important instructions:

- Responses are \*\*ordinal\*\*: higher numbers indicate stronger agreement, endorsement, or intensity (as implied by the question).

- Output a probability distribution over responses {1,2,3,4}.

- The distribution should reflect realistic human uncertainty: do NOT assume the persona always responds deterministically.

- If the persona strongly aligns with one side, assign higher probability there, but still allow nonzero probability for nearby options.

- The probabilities must be non-negative and sum to exactly 1.

- Avoid assigning probability 1.0 or 0.0 unless the persona makes all other responses essentially impossible.

Output format: Return ONLY a JSON-style list of four numbers: [p1, p2, p3, p4]. Do not include any explanation or additional text.

---

---

**User Prompt**

PERSONA PROFILE: {persona}
SURVEY QUESTION: {question}
FORMAT INSTRUCTIONS: Return ONLY a JSON-style list of four numbers: [p1, p2, p3, p4]. Do not include any explanation or additional text.

---

## E. Additional Results

### E.1. Additional Results on WorldValuesBench

This appendix collects additional numerical results that complement the log-loss curves shown in Figure 2 of the main text. Tables 3 and 4 report log-loss values at representative budgets for synthetic and real users, respectively, and serve as the tabular counterparts of Figure 2. Figures 3 and 4 and Tables 5–8 further report Brier score and ordinal MSE for the same experiments. In all plots, curves denote means averaged over test user–target-question pairs, and shaded regions correspond to 95% confidence intervals.

*Table 3.* Synthetic users: log loss by query budget $T$. $N = 20{,}000$ test users; cells report mean with standard error below. At $T = 86$ all feasible questions have been asked, so all persona-based methods coincide with the **full** baseline. **Bold** marks the best value per column.

| Method | $T\!=\!5$ | $T\!=\!10$ | $T\!=\!15$ | $T\!=\!20$ | $T\!=\!30$ | $T\!=\!50$ | $T\!=\!86$ |
|---|---|---|---|---|---|---|---|
| random | 1.115 (.002) | 1.097 (.002) | 1.084 (.002) | 1.074 (.002) | 1.059 (.002) | 1.034 (.002) | **1.006** (.002) |
| random_fixed | 1.109 (.002) | 1.095 (.002) | 1.077 (.002) | 1.070 (.002) | 1.056 (.002) | 1.031 (.002) | **1.006** (.002) |
| nonadaptive | 1.104 (.002) | 1.094 (.002) | 1.080 (.002) | 1.074 (.002) | 1.057 (.002) | 1.029 (.002) | **1.006** (.002) |
| greedy | **1.089** (.002) | **1.068** (.002) | **1.055** (.002) | **1.043** (.002) | **1.026** (.002) | **1.011** (.002) | **1.006** (.002) |
| CAT-GRM | 1.207 (.003) | 1.196 (.002) | 1.181 (.002) | 1.167 (.002) | 1.154 (.002) | 1.146 (.002) | 1.146 (.002) |
| CAT-GPCM | 1.338 (.004) | 1.318 (.004) | 1.286 (.004) | 1.265 (.004) | 1.231 (.003) | 1.199 (.003) | 1.194 (.003) |
| CAT-MGRM | 1.164 (.002) | 1.157 (.002) | 1.156 (.002) | 1.152 (.002) | 1.149 (.002) | 1.145 (.002) | 1.141 (.002) |
| CAT-MGPCM | 1.283 (.003) | 1.225 (.003) | 1.206 (.003) | 1.199 (.003) | 1.187 (.003) | 1.190 (.003) | 1.186 (.003) |

*Table 4.* Real users (WorldValuesBench): log loss by query budget $T$ for all methods. $N = 17{,}692$ held-out users; cells report mean with standard error below. At $T = 86$ all feasible questions have been asked, so all persona-based methods coincide with the **full** baseline. **Bold** marks the best value per column.

| Method | $T\!=\!5$ | $T\!=\!10$ | $T\!=\!15$ | $T\!=\!20$ | $T\!=\!30$ | $T\!=\!50$ | $T\!=\!86$ |
|---|---|---|---|---|---|---|---|
| random | 1.033 (.002) | 1.022 (.002) | 1.015 (.002) | 1.010 (.002) | 1.005 (.002) | 1.000 (.002) | **.998** (.002) |
| random_fixed | 1.040 (.002) | 1.028 (.002) | 1.036 (.002) | 1.036 (.002) | 1.037 (.002) | 1.012 (.002) | **.998** (.002) |
| nonadaptive | 1.039 (.002) | 1.020 (.002) | **.991** (.002) | **.977** (.002) | **.981** (.002) | **.988** (.002) | **.998** (.002) |
| greedy | **1.019** (.002) | **1.006** (.002) | 1.001 (.002) | .999 (.002) | .997 (.002) | .998 (.002) | **.998** (.002) |
| CAT-GRM | 1.074 (.003) | 1.067 (.003) | 1.058 (.003) | 1.046 (.003) | 1.031 (.003) | 1.019 (.003) | 1.005 (.003) |
| CAT-GPCM | 1.149 (.004) | 1.122 (.004) | 1.104 (.004) | 1.092 (.003) | 1.067 (.003) | 1.048 (.003) | 1.033 (.003) |
| CAT-MGRM | 1.044 (.003) | 1.033 (.003) | 1.022 (.003) | 1.015 (.003) | 1.011 (.003) | 1.011 (.003) | 1.006 (.003) |
| CAT-MGPCM | 1.107 (.004) | 1.083 (.003) | 1.071 (.003) | 1.066 (.003) | 1.051 (.003) | 1.038 (.003) | 1.032 (.003) |

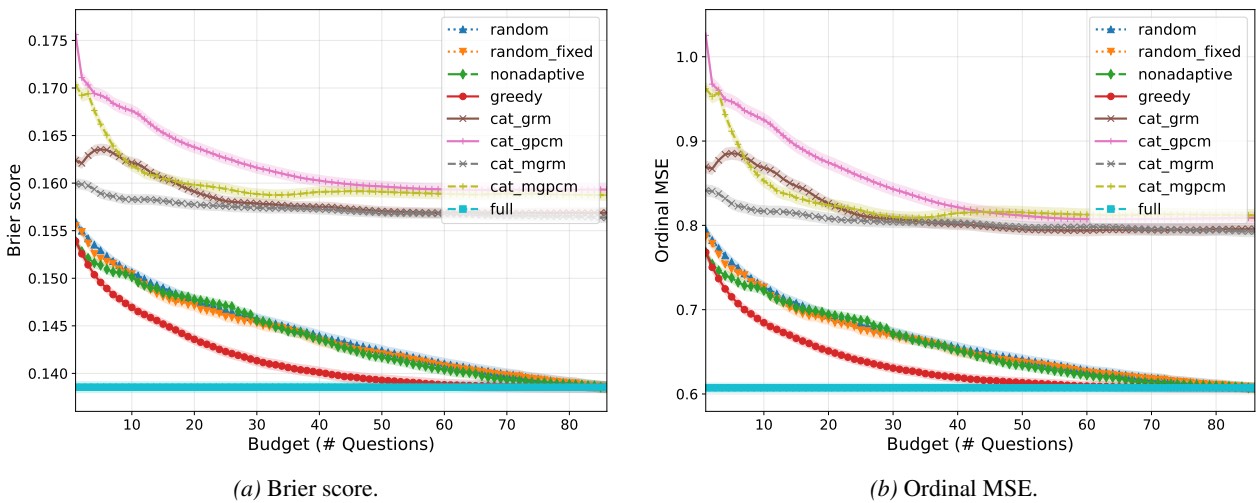

*(a)* Brier score.

*(b)* Ordinal MSE.

*Figure 3.* Synthetic users: performance of all methods as a function of query budget, evaluated using Brier score and ordinal MSE. Persona-based methods substantially outperform CAT baselines, with greedy achieving the strongest performance among persona-based approaches.

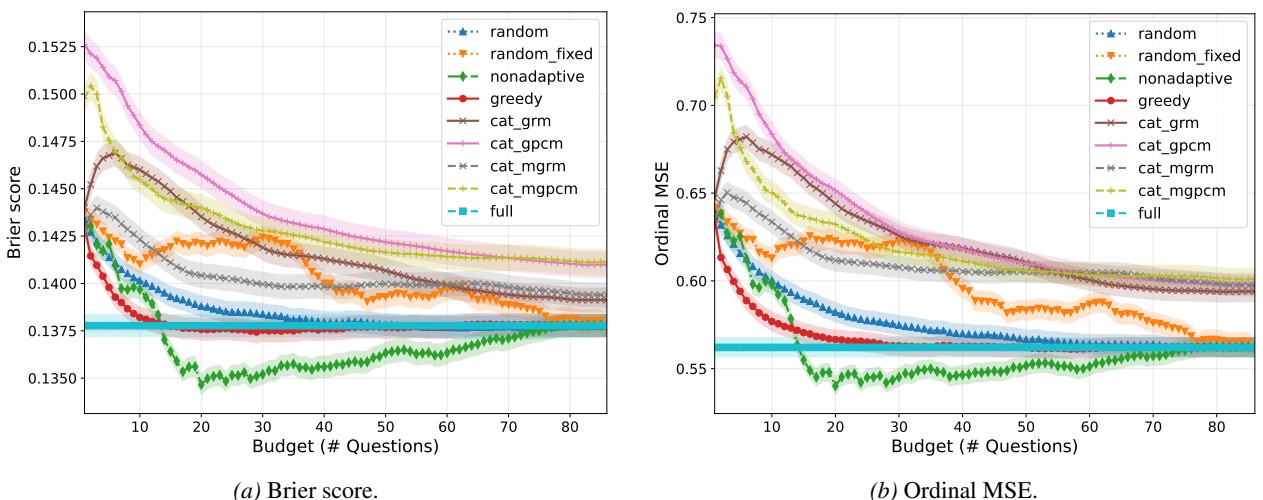

*(a)* Brier score.

*(b)* Ordinal MSE.

*Figure 4.* Real users: performance of all methods as a function of query budget, evaluated using Brier score and ordinal MSE. Persona-based methods outperform CAT baselines; greedy performs best at small budgets, while non-adaptive designs can overtake at larger budgets.

Tables 5 and 6 report per-budget Brier score and ordinal MSE for synthetic users, complementing the log-loss results in Table 3. Tables 7 and 8 report the per-budget Brier score and ordinal MSE values for real users, complementing the log-loss results in Table 4.

*Table 5.* Synthetic users: Brier score by query budget $T$. $N = 20{,}000$ test users; cells report mean with standard error below. At $T = 86$ all feasible questions have been asked, so all persona-based methods coincide with the **full** baseline. **Bold** marks the best value per column.

| Method | $T=5$ | $T=10$ | $T=15$ | $T=20$ | $T=30$ | $T=50$ | $T=86$ |
|---|---|---|---|---|---|---|---|
| random | .1529 (.0003) | .1506 (.0003) | .1490 (.0003) | .1477 (.0003) | .1458 (.0003) | .1424 (.0003) | **.1386** (.0003) |
| random_fixed | .1521 (.0003) | .1504 (.0003) | .1481 (.0003) | .1471 (.0003) | .1453 (.0003) | .1420 (.0003) | **.1386** (.0003) |
| nonadaptive | .1514 (.0003) | .1501 (.0003) | .1485 (.0003) | .1478 (.0003) | .1456 (.0003) | .1417 (.0003) | **.1386** (.0003) |
| greedy | **.1496** (.0003) | **.1469** (.0003) | **.1452** (.0003) | **.1436** (.0003) | **.1413** (.0003) | **.1393** (.0003) | **.1386** (.0003) |
| CAT-GRM | .1635 (.0003) | .1621 (.0003) | .1605 (.0003) | .1591 (.0003) | .1579 (.0003) | .1570 (.0003) | .1569 (.0003) |
| CAT-GPCM | .1692 (.0004) | .1676 (.0003) | .1653 (.0003) | .1638 (.0003) | .1616 (.0003) | .1597 (.0003) | .1593 (.0003) |
| CAT-MGRM | .1589 (.0003) | .1583 (.0003) | .1581 (.0003) | .1578 (.0003) | .1574 (.0003) | .1569 (.0003) | .1564 (.0003) |
| CAT-MGPCM | .1662 (.0003) | .1618 (.0003) | .1605 (.0003) | .1598 (.0003) | .1589 (.0003) | .1591 (.0003) | .1587 (.0003) |

*Table 6.* Synthetic users: ordinal MSE by query budget $T$. $N = 20{,}000$ test users; cells report mean with standard error below. At $T = 86$ all feasible questions have been asked, so all persona-based methods coincide with the **full** baseline. **Bold** marks the best value per column.

| Method | $T=5$ | $T=10$ | $T=15$ | $T=20$ | $T=30$ | $T=50$ | $T=86$ |
|---|---|---|---|---|---|---|---|
| random | .757 (.003) | .728 (.003) | .708 (.003) | .693 (.003) | .673 (.003) | .641 (.003) | **.607** (.003) |
| random_fixed | .748 (.003) | .727 (.003) | .698 (.003) | .688 (.003) | .669 (.003) | .637 (.003) | **.607** (.003) |
| nonadaptive | .738 (.003) | .723 (.003) | .703 (.003) | .694 (.003) | .671 (.003) | .634 (.003) | **.607** (.003) |
| greedy | **.715** (.003) | **.684** (.003) | **.666** (.003) | **.651** (.003) | **.631** (.003) | **.614** (.003) | **.607** (.003) |
| CAT-GRM | .886 (.004) | .868 (.004) | .847 (.004) | .826 (.004) | .807 (.004) | .795 (.004) | .796 (.004) |
| CAT-GPCM | .947 (.005) | .925 (.004) | .895 (.004) | .874 (.004) | .843 (.004) | .812 (.004) | .809 (.004) |
| CAT-MGRM | .826 (.004) | .817 (.004) | .814 (.004) | .808 (.004) | .804 (.004) | .798 (.004) | .793 (.004) |
| CAT-MGPCM | .912 (.004) | .853 (.004) | .833 (.004) | .824 (.004) | .810 (.004) | .815 (.004) | .812 (.004) |

*Table 7.* Real users (WorldValuesBench): Brier score by query budget $T$. $N = 17{,}692$ held-out users; cells report mean with standard error below. At $T = 86$ all feasible questions have been asked, so all persona-based methods coincide with the **full** baseline. **Bold** marks the best value per column.

| Method | $T=5$ | $T=10$ | $T=15$ | $T=20$ | $T=30$ | $T=50$ | $T=86$ |
|---|---|---|---|---|---|---|---|
| random | .1414 (.0003) | .1402 (.0003) | .1393 (.0003) | .1388 (.0003) | .1384 (.0003) | .1378 (.0003) | **.1378** (.0003) |
| random_fixed | .1424 (.0003) | .1410 (.0003) | .1419 (.0003) | .1421 (.0003) | .1424 (.0003) | .1393 (.0003) | **.1378** (.0003) |
| nonadaptive | .1421 (.0003) | .1397 (.0003) | **.1364** (.0003) | **.1347** (.0003) | **.1352** (.0003) | **.1363** (.0003) | **.1378** (.0003) |
| greedy | **.1398** (.0003) | **.1382** (.0003) | .1377 (.0003) | .1376 (.0003) | .1375 (.0003) | .1377 (.0003) | **.1378** (.0003) |
| CAT-GRM | .1467 (.0004) | .1460 (.0004) | .1449 (.0004) | .1435 (.0004) | .1419 (.0003) | .1407 (.0003) | .1391 (.0003) |
| CAT-GPCM | .1509 (.0004) | .1484 (.0004) | .1468 (.0004) | .1457 (.0004) | .1437 (.0004) | .1422 (.0004) | .1410 (.0004) |
| CAT-MGRM | .1436 (.0003) | .1423 (.0003) | .1411 (.0003) | .1404 (.0003) | .1400 (.0003) | .1400 (.0003) | .1394 (.0003) |
| CAT-MGPCM | .1476 (.0004) | .1455 (.0004) | .1444 (.0004) | .1440 (.0004) | .1428 (.0004) | .1416 (.0004) | .1412 (.0004) |

*Table 8.* Real users (WorldValuesBench): ordinal MSE by query budget $T$. $N = 17{,}692$ held-out users; cells report mean with standard error below. At $T = 86$ all feasible questions have been asked, so all persona-based methods coincide with the **full** baseline. **Bold** marks the best value per column.

| Method | $T=5$ | $T=10$ | $T=15$ | $T=20$ | $T=30$ | $T=50$ | $T=86$ |
|---|---|---|---|---|---|---|---|
| random | .616 (.003) | .600 (.003) | .590 (.003) | .582 (.003) | .575 (.003) | .567 (.003) | **.562** (.003) |
| random_fixed | .627 (.003) | .613 (.003) | .623 (.003) | .623 (.003) | .623 (.003) | .583 (.003) | **.562** (.003) |
| nonadaptive | .625 (.003) | .598 (.003) | **.555** (.003) | **.540** (.003) | **.545** (.003) | **.552** (.003) | **.562** (.003) |
| greedy | **.594** (.003) | **.577** (.003) | .571 (.003) | .567 (.003) | .563 (.003) | .562 (.003) | **.562** (.003) |
| CAT-GRM | .681 (.003) | .672 (.003) | .659 (.003) | .644 (.003) | .626 (.003) | .610 (.003) | .594 (.003) |
| CAT-GPCM | .714 (.004) | .684 (.004) | .663 (.004) | .651 (.003) | .628 (.003) | .610 (.003) | .598 (.003) |
| CAT-MGRM | .646 (.003) | .634 (.003) | .620 (.003) | .612 (.003) | .608 (.003) | .605 (.003) | .597 (.003) |
| CAT-MGPCM | .676 (.004) | .650 (.003) | .637 (.003) | .632 (.003) | .616 (.003) | .606 (.003) | .601 (.003) |

## E.2. Ablation Studies

This appendix provides full results and detailed protocols for the three ablation axes summarized in Section 4.5. Tables 9 and 10 report log loss for the **nonadaptive** and **greedy** methods, respectively, on $N = 10{,}000$ sampled real users. Brier score and ordinal MSE show the same qualitative patterns and are omitted for brevity.

**Persona dictionary clustering.** The full Twin-2K-500 dictionary contains $n = 2{,}058$ personas, which may be larger than necessary for effective inference. To assess sensitivity to dictionary size, we compress the dictionary into a smaller set of prototype personas. Concretely, we first prune low-mass personas using the empirical Bayes prior learned from training users, then cluster the remaining personas with prior-weighted $k$-means using Jensen–Shannon divergence as the distance metric, and construct each prototype as the prior-weighted average of member personas' response distributions across questions. The prior over prototypes is set to the sum of priors of personas assigned to each cluster, preserving Bayesian consistency in the reduced dictionary. Tables 9 and 10 show results for $n \in \{50, 200\}$ clusters for non-adaptive and greedy methods, respectively. Performance is robust down to approximately 200 clusters, with limited degradation at 50 clusters. At 200 clusters, the greedy method even slightly improves at some budgets, likely because pruning redundant or noisy personas reduces posterior diffusion. This suggests that moderate compression can reduce computational cost with minimal loss in

predictive accuracy.

**Deterministic-with-noise responses.** A natural question is whether the full distributional shape of the LLM-elicited response probabilities matters, or whether a simpler point-prediction approach suffices. To test this, we replace the elicited distributions with a deterministic (mode) response plus uniform noise: for each persona–question pair, we first prompt the LLM to output a single canonical answer $\hat{y}$ (the most likely response option), and then define a noisy categorical distribution $p(y) = (1 - \varepsilon)\mathbf{1}\{y = \hat{y}\} + \varepsilon/(K - 1)$, where $\varepsilon \in \{0.1, 0.3\}$ controls the sharpness of the persona model. This ablation uniformly degrades performance across all methods and budgets—often dramatically so (e.g., log loss exceeding 1.3 at moderate budgets for $\varepsilon = 0.1$). The degradation is especially severe for small $\varepsilon$, where the near-deterministic likelihoods cause the posterior to concentrate rapidly on a single persona, leaving little room for correction after early misassignment. This confirms that the distributional shape of the LLM-elicited responses carries substantial information beyond the modal answer, and that directly eliciting probability distributions from the LLM is a meaningfully better strategy than eliciting point predictions and injecting synthetic noise.

**Temperature scaling.** We apply temperature scaling to the LLM-elicited distributions, raising probabilities to a power of $1/\tau$ and re-normalizing: $\hat{p}_\tau(y = k) \propto \hat{p}(y = k)^{1/\tau}$, where $\tau = 1$ recovers the original distribution, $\tau < 1$ sharpens it, and $\tau > 1$ softens it. Results for $\tau \in \{0.5, 2\}$ show that both sharpening and softening consistently degrade performance. Sharpening ($\tau = 0.5$) can initially appear competitive at very small budgets for the nonadaptive method, but degrades sharply as budget increases due to overconfident likelihoods that amplify posterior misassignment. Softening ($\tau = 2$) uniformly underperforms by washing out the discriminative signal in the response distributions. These results suggest that the original LLM-elicited distributions are already well-calibrated for the adaptive querying objective, and that post-hoc rescaling is unlikely to improve performance without additional task-specific calibration data.

*Table 9.* Ablation study: log loss for **nonadaptive** design on real users ($N = 10,000$ sampled); cells report mean with standard error below. **Bold** marks the best value per column.

| Variant | $T=5$ | $T=10$ | $T=15$ | $T=20$ | $T=30$ | $T=50$ | $T=86$ |
|---|---|---|---|---|---|---|---|
| current setup | 1.022 (.006) | 1.000 (.006) | **.976** (.006) | **.962** (.006) | **.961** (.006) | **.968** (.006) | .979 (.006) |
| cluster = 50 | 1.023 (.006) | 1.005 (.006) | .986 (.006) | .977 (.006) | .970 (.006) | .970 (.006) | .977 (.006) |
| cluster = 200 | 1.022 (.006) | .997 (.006) | .977 (.006) | .963 (.006) | .962 (.006) | **.968** (.006) | **.976** (.006) |
| det. $\varepsilon = 0.1$ | 1.029 (.010) | 1.199 (.012) | 1.272 (.013) | 1.291 (.014) | 1.351 (.014) | 1.421 (.015) | 1.472 (.015) |
| det. $\varepsilon = 0.3$ | 1.080 (.006) | 1.064 (.007) | 1.081 (.007) | 1.106 (.008) | 1.116 (.008) | 1.136 (.008) | 1.168 (.009) |
| temp $\tau = 0.5$ | **1.005** (.009) | **.981** (.009) | 1.022 (.010) | 1.021 (.010) | 1.034 (.011) | 1.071 (.011) | 1.085 (.011) |
| temp $\tau = 2$ | 1.097 (.004) | 1.087 (.004) | 1.075 (.004) | 1.074 (.004) | 1.071 (.004) | 1.068 (.004) | 1.069 (.004) |

# F. CAT Baselines

This appendix provides a self-contained overview of computerized adaptive testing (CAT) and the item response theory (IRT) models used as baselines in our experiments. We describe (i) the response models, (ii) parameter estimation via marginal maximum likelihood (MML), (iii) posterior updates during adaptive testing, and (iv) item selection criteria. Throughout, responses take values in $\{0, 1, \ldots, K - 1\}$.

## F.1. Overview of Computerized Adaptive Testing

A classical CAT system consists of three components:

1. **Item response model:** a probabilistic model $P(Y_x = k \mid \theta)$ describing how a latent trait $\theta$ (or $\boldsymbol{\theta}$) governs responses to item $x$.

2. **Posterior inference:** an update rule for the posterior distribution $p(\theta \mid \mathcal{D}_t)$ given observed item–response pairs

*Table 10.* Ablation study: log loss for **greedy** design on real users ($N = 10{,}000$ sampled); cells report mean with standard error below. **Bold** marks the best value per column.

| Variant | $T\!=\!5$ | $T\!=\!10$ | $T\!=\!15$ | $T\!=\!20$ | $T\!=\!30$ | $T\!=\!50$ | $T\!=\!86$ |
|---|---|---|---|---|---|---|---|
| current setup | **1.003** (.006) | .993 (.006) | .987 (.006) | **.984** (.006) | **.984** (.006) | .984 (.006) | .979 (.006) |
| cluster = 50 | 1.012 (.006) | .999 (.006) | .996 (.006) | .990 (.006) | .985 (.006) | .981 (.006) | .977 (.006) |
| cluster = 200 | 1.004 (.006) | **.992** (.006) | **.986** (.006) | .986 (.006) | .986 (.006) | **.978** (.006) | **.976** (.006) |
| det. $\varepsilon = 0.1$ | 1.218 (.012) | 1.329 (.013) | 1.368 (.014) | 1.399 (.014) | 1.427 (.015) | 1.490 (.015) | 1.472 (.015) |
| det. $\varepsilon = 0.3$ | 1.065 (.007) | 1.085 (.007) | 1.106 (.008) | 1.121 (.008) | 1.138 (.008) | 1.167 (.009) | 1.168 (.009) |
| temp $\tau = 0.5$ | 1.052 (.010) | 1.078 (.011) | 1.097 (.011) | 1.094 (.012) | 1.107 (.012) | 1.101 (.012) | 1.085 (.011) |
| temp $\tau = 2$ | 1.098 (.004) | 1.088 (.004) | 1.082 (.004) | 1.078 (.004) | 1.075 (.004) | 1.072 (.004) | 1.069 (.004) |

$$\mathcal{D}_t = \{(x_1, y_1), \ldots, (x_t, y_t)\}.$$

3. **Item selection:** a criterion for selecting the next item $x_{t+1}$ to efficiently reduce uncertainty about $\theta$.

In cognitive testing, $\theta$ typically represents ability. In our survey prediction setting, $\theta$ captures latent attitudes or factors that shape responses. After $T$ adaptive queries, predictions for unasked items are made via the posterior predictive distribution

$$P(Y_x = k \mid \mathcal{D}_T) = \int P(Y_x = k \mid \theta)\, p(\theta \mid \mathcal{D}_T)\, d\theta,$$

or its multidimensional analogue.

### F.2. Item Response Theory Models

We implement four IRT models: two unidimensional polytomous models (GRM, GPCM) and their multidimensional extensions (MGRM, MGPCM). All models assume conditional independence across items given the latent trait.

#### F.2.1. GRADED RESPONSE MODEL (GRM)

The graded response model (GRM) is a cumulative-link model for ordinal responses. For item $x$ with ordered categories $\{0, 1, \ldots, K-1\}$, GRM defines cumulative probabilities

$$P(Y_x \geq k \mid \theta) = \sigma(a_x(\theta - b_{x,k})), \qquad k = 1, \ldots, K-1, \tag{12}$$

where $\sigma(u) = 1/(1 + e^{-u})$ and $a_x > 0$ is the discrimination parameter. The thresholds satisfy $b_{x,1} \leq b_{x,2} \leq \cdots \leq b_{x,K-1}$. With conventions $P(Y_x \geq 0 \mid \theta) = 1$ and $P(Y_x \geq K \mid \theta) = 0$, category probabilities are obtained by differencing:

$$P(Y_x = k \mid \theta) = P(Y_x \geq k \mid \theta) - P(Y_x \geq k+1 \mid \theta), \qquad k = 0, \ldots, K-1. \tag{13}$$

#### F.2.2. GENERALIZED PARTIAL CREDIT MODEL (GPCM)

The generalized partial credit model (GPCM) is an adjacent-category model. One convenient parameterization yields the softmax form

$$P(Y_x = k \mid \theta) = \frac{\exp\!\left(\sum_{s=1}^{k} a_x(\theta - d_{x,s})\right)}{\sum_{m=0}^{K-1} \exp(\sum_{s=1}^{m} a_x(\theta - d_{x,s}))}, \qquad k = 0, \ldots, K-1, \tag{14}$$

where $a_x > 0$ is discrimination and $\{d_{x,s}\}_{s=1}^{K-1}$ are step parameters (with the convention that an empty sum equals 0). Unlike GRM, GPCM does not require ordered thresholds.

### F.2.3. MULTIDIMENSIONAL GRM (MGRM)

The multidimensional GRM replaces the scalar trait with $\boldsymbol{\theta} \in \mathbb{R}^D$ and uses an item discrimination vector $\mathbf{a}_x \in \mathbb{R}^D$:

$$P(Y_x \geq k \mid \boldsymbol{\theta}) = \sigma\left(\mathbf{a}_x^\top \boldsymbol{\theta} - b_{x,k}\right), \qquad k = 1, \ldots, K-1, \tag{15}$$

with category probabilities again computed by differencing consecutive cumulative probabilities.

### F.2.4. MULTIDIMENSIONAL GPCM (MGPCM)

Similarly, the multidimensional GPCM uses $\mathbf{a}_x \in \mathbb{R}^D$ and step parameters $\{d_{x,s}\}$:

$$P(Y_x = k \mid \boldsymbol{\theta}) = \frac{\exp\left(\sum_{s=1}^{k} \mathbf{a}_x^\top \boldsymbol{\theta} - d_{x,s}\right)}{\sum_{m=0}^{K-1} \exp(\sum_{s=1}^{m} \mathbf{a}_x^\top \boldsymbol{\theta} - d_{x,s})}, \qquad k = 0, \ldots, K-1, \tag{16}$$

again using the empty-sum convention. This form reduces to (14) in the unidimensional case.

### F.3. Parameter Estimation via MML (EM)

IRT parameters are fitted on training users using marginal maximum likelihood (MML). Let $Y^{(i)}$ denote the responses for user $i$, with missing entries omitted from the product below. Under conditional independence,

$$p(Y^{(i)} \mid \theta) = \prod_{x \in \mathcal{I}_{\text{obs}}^{(i)}} P(Y_x^{(i)} \mid \theta),$$

where $\mathcal{I}_{\text{obs}}^{(i)}$ is the set of observed items for user $i$. MML maximizes the marginal log-likelihood

$$\sum_{i=1}^{N} \log \int p(Y^{(i)} \mid \theta) \, \phi(\theta) \, d\theta$$

(or the multidimensional analogue), where $\phi$ denotes the standard normal prior.

#### F.3.1. GRID-BASED APPROXIMATION

We approximate integrals over $\theta$ using a fixed grid. For unidimensional models, we discretize $\theta \in [-\theta_{\max}, \theta_{\max}]$ using $G$ grid points $\{\theta^{(g)}\}_{g=1}^{G}$ with weights $w^{(g)} \propto \phi(\theta^{(g)})$. For multidimensional models with $D$ dimensions, we use a Cartesian grid with $G$ points per dimension, yielding $G^D$ grid points $\{\boldsymbol{\theta}^{(g)}\}_{g=1}^{G^D}$ with weights proportional to the multivariate normal density.

#### F.3.2. EM UPDATES

Given current item parameters, the E-step computes responsibilities

$$\pi_{i,g} = \frac{w^{(g)} \prod_{x \in \mathcal{I}_{\text{obs}}^{(i)}} P(Y_x^{(i)} \mid \theta^{(g)})}{\sum_{g'} w^{(g')} \prod_{x \in \mathcal{I}_{\text{obs}}^{(i)}} P(Y_x^{(i)} \mid \theta^{(g')})}. \tag{17}$$

The M-step updates item parameters by maximizing the expected complete-data log-likelihood, separately for each item $x$:

$$\text{params}(x) = \operatorname{argmax} \sum_{i:\, x \in \mathcal{I}_{\text{obs}}^{(i)}} \sum_g \pi_{i,g} \, \log P(Y_x^{(i)} \mid \theta^{(g)}; \text{params}(x)). \tag{18}$$

For GRM we enforce $a_x > 0$ and ordered thresholds $b_{x,1} \leq \cdots \leq b_{x,K-1}$; optimization is performed with L-BFGS-B.

### F.4. Posterior Updates During Adaptive Testing

During adaptive testing, we maintain the posterior over the latent trait on the same grid. Let $w_t^{(g)}$ denote the posterior weight on grid point $\theta^{(g)}$ after observing $\mathcal{D}_t$.

**Initialization.** We initialize $w_0^{(g)} \propto \phi(\theta^{(g)})$ (or multivariate normal for MIRT), normalized to sum to $1$.

**Bayesian update.** After querying item $x_t$ and observing response $y_t$, the posterior weights update as

$$w_t^{(g)} = \frac{w_{t-1}^{(g)} P(Y_{x_t} = y_t \mid \theta^{(g)})}{\sum_{g'} w_{t-1}^{(g')} P(Y_{x_t} = y_t \mid \theta^{(g')})}. \tag{19}$$

**Posterior summaries.** For 1D models, we compute the posterior mean and variance via

$$\hat{\theta}_t = \sum_g w_t^{(g)} \theta^{(g)}, \qquad \mathrm{Var}(\theta \mid \mathcal{D}_t) = \sum_g w_t^{(g)} (\theta^{(g)} - \hat{\theta}_t)^2,$$

and similarly compute posterior covariance for multidimensional models.

### F.5. Item Selection Criteria

We implement standard CAT selection rules. Let $\mathcal{I}_t$ denote the set of items already administered to the user.

#### F.5.1. MAXIMUM FISHER INFORMATION (MFI)

MFI selects the next item by maximizing Fisher information at a point estimate (typically $\hat{\theta}_t$):

$$x_{t+1} = \operatorname*{argmax}_{x \in \mathcal{I}_{\mathrm{feas}} \setminus \mathcal{I}_t} I_x(\hat{\theta}_t). \tag{20}$$

For polytomous models, Fisher information can be written as

$$I_x(\theta) = \sum_{k=0}^{K-1} \frac{\left(\frac{\partial}{\partial \theta} P(Y_x = k \mid \theta)\right)^2}{P(Y_x = k \mid \theta)}. \tag{21}$$

MFI is computationally efficient but uses only a point estimate rather than the full posterior.

#### F.5.2. MINIMUM EXPECTED POSTERIOR VARIANCE (MEPV)

MEPV is a Bayesian criterion that selects the item minimizing expected posterior variance after observing the (unknown) response:

$$x_{t+1} = \operatorname*{argmin}_{x \in \mathcal{I}_{\mathrm{feas}} \setminus \mathcal{I}_t} \mathbb{E}_{Y_x \mid \mathcal{D}_t}[\mathrm{Var}(\theta \mid \mathcal{D}_t, Y_x)]. \tag{22}$$

The expectation is computed under the posterior predictive distribution

$$P(Y_x = k \mid \mathcal{D}_t) = \sum_g w_t^{(g)} P(Y_x = k \mid \theta^{(g)}). \tag{23}$$

For each possible response $k$, we form the hypothetical updated posterior via (19), compute its variance, and average over $k$. In our experiments, we use MEPV for 1D baselines.

#### F.5.3. MULTIDIMENSIONAL CRITERIA

For $D$-dimensional traits, Fisher information becomes a matrix $\mathbf{I}_x(\boldsymbol{\theta})$. We use an A-optimality-style Bayesian criterion that minimizes the expected trace of the posterior covariance matrix:

$$x_{t+1} = \operatorname*{argmin}_{x \in \mathcal{I}_{\mathrm{feas}} \setminus \mathcal{I}_t} \mathbb{E}_{Y_x \mid \mathcal{D}_t}[\mathrm{tr}(\boldsymbol{\Sigma}_{t+1})],$$

which reduces to MEPV when $D = 1$.

### F.6. Prediction

After $T$ adaptive queries, predictions for any target item $x$ use the posterior predictive distribution on the grid:

$$P(Y_x = k \mid \mathcal{D}_T) = \sum_g w_T^{(g)} P(Y_x = k \mid \theta^{(g)}). \tag{24}$$

### F.7. Implementation Details

Table 11 summarizes hyperparameters used in our CAT implementations.

*Table 11.* Hyperparameters for CAT baseline implementations.

| Parameter | Description | Value |
|---|---|---|
| *Grid-based posterior* | | |
| $\theta_{\max}$ | Grid range: $\theta \in [-\theta_{\max}, \theta_{\max}]$ | 4.0 |
| $G$ (1D) | Number of grid points for GRM/GPCM | 41 |
| $G$ (MIRT) | Grid points per dimension for MGRM/MGPCM | 9 |
| $D$ | Latent dimensions for MIRT models | 3 |
| *Parameter estimation (EM)* | | |
| Max iterations | Maximum EM iterations | 50 |
| Tolerance | Convergence criterion (log-likelihood change) | $10^{-3}$ |
| *Item selection* | | |
| Criterion (1D) | Selection criterion for GRM/GPCM | MEPV |
| Criterion (MIRT) | Selection criterion for MGRM/MGPCM | A-optimality |

