# OpenReview forum: "Adaptive Querying with AI Persona Priors"
_ICML.cc/2026/Conference — ICML 2026 regular_

### Official Review · Reviewer_Wvbc · 2026-03-10

**Soundness:** 4
**Presentation:** 4
**Significance:** 3
**Originality:** 3
**Overall Recommendation:** 5
**Confidence:** 5

**Summary:**

The paper proposes a novel framework for Bayesian adaptive querying that addresses the computational bottlenecks and cold-start limitations of traditional interactive systems.  The authors introduce a latent variable model where the user's state is represented as a finite mixture over a dictionary of AI personas.  A LLM is prompted to generate the response distributions for each persona-question pair, forming an explicit probabilistic prior within the bayesian inference framework.  This finite-mixture structure allows for closed-form posterior updates and efficient predictive inference, effectively turning the complex problem of Bayesian experimental design into a highly tractable and scalable procedure.  By mapping complex user heterogeneity to a discrete set of AI personas with categorical responses, the model entirely bypasses the need for computationally expensive posterior approximations like nested Monte Carlo or variational inference.  Additionally, the approach leverages the vast pretraining knowledge of LLMs to generate structure prior beliefs.  This allow the system to be effective even when little to no historical calibration data is available, an area where traditional models struggle.  Unlike many recent works that treat LLMs as heuristic black boxes for elicitation, this paper formalizes the LLM's output as a strict probabilistic prior within a mathematically rigorous Bayesian framework.

**Compliance With Llm Reviewing Policy:**

Affirmed.

**Key Questions For Authors:**

None

**Limitations:**

Yes

**Strengths And Weaknesses:**

The authors developed their approach from the rigorous theory of Bayesian experimental design.  The topic is extensively researched by the authors with the presentation of the shortcomings of prior approaches and survey of existing methodologies and solution methods.  The presentation is clear, well structured and easy to follow.  The novelty of modeling the LLM response as a structured prior is refreshing and provides new insights for a disciplined approach to adaptive testing using LLMs.  The formal Bayesian framework provides a theory grounded method of using LLMs more effectively in problem design.  They also make an insightful observation regarding model misspecification on real users.

---

> ### Author Rebuttal · Authors · 2026-03-30
>
> We sincerely thank the reviewer for the very positive assessment and encouraging comments. We are especially glad that the reviewer found the combination of Bayesian experimental design and LLM-derived persona priors both principled and practically useful. We will use the revision to further improve clarity and better highlight these strengths.

---

> > ### Author Rebuttal · Reviewer_Wvbc · 2026-03-31
> >
> > Nothing to be resolved.

---

### Official Review · Reviewer_yZoh · 2026-03-12

**Soundness:** 3
**Presentation:** 4
**Significance:** 4
**Originality:** 4
**Overall Recommendation:** 4
**Confidence:** 3

**Summary:**

This paper proposes adaptive querying with AI persona priors. Differ from classical Bayesian framework, it introduces a persona-induced latent variable model.
Offline, the authors collect persona-question response distribution from an LLM for a dictionary of personas. Online, a new user’s posterior is updated through Bayesian adaptive querying.
It experiments on synthetic data (well-specified) and WorldValuesBench (misspecified), demonstrates that persona-based posteriors deliver accurate probabilistic prediction, and outperform classical CAT/IRT baseline.

**Compliance With Llm Reviewing Policy:**

Affirmed.

**Key Questions For Authors:**

1. Although referenced, could you explain your persona dictionary, such as sampling diversity, question domain, in the main text?
2. For figure 3, the authors claim that nonadaptive outperforms greedy because of overfit, could you provide a deeper diagnosis?
3. Could you provide runtime comparisons?
4. Did you consider latent class analysis (LCA) as a baseline?

**Limitations:**

yes

**Strengths And Weaknesses:**

Strength
It introduces a persona-induced latent variable model that yields expressive prior over high-dimensional response;
It provides tractable Bayesian inference, avoiding nested Monte Carlo or variational approximation;
It evaluates both on synthetic data and real data (WorldValuesBench), and in both cases outperform baselines;
It provides several ablation studies

Weakness
It assumes conditional independence across questions given persona;
The persona dictionary is fixed, and doesn’t explore the result sensitivity to the dictionary;
Some clarity questions (asked in question section)

---

> ### Author Rebuttal · Authors · 2026-03-30
>
> We thank the reviewer for the positive assessment and helpful questions.
>
> > **Conditional independence across questions given persona may be restrictive.**
>
> We agree that this is a simplifying assumption. We adopt it for the same reason many latent-variable models (like CAT) do: it keeps posterior updates and predictive inference tractable. In our case, it also makes the offline construction of persona-question likelihoods scalable. We will clarify this assumption and its limitations more explicitly in the revision.
>
> > **Please explain the persona dictionary more clearly in the main text (sampling diversity, question domain, etc.).**
>
> We will include more details of the persona dictionary in the main text. In short, the persona bank comes from **Twin-2K-500**, with 2,058 survey-based persona profiles derived from a heterogeneous sample of U.S. participants covering more than 500 questions across demographic, psychological, economic, and behavioral domains. This gives the dictionary three desirable properties: **diversity** across user types, **broad domain coverage**, and **grounding in real human profiles** rather than hand-crafted personas.
>
> > **Can you give a deeper diagnosis of why `nonadaptive` can outperform `greedy` in Figure 3?**
>
> Our current diagnosis is a **misspecification-robustness tradeoff**. `greedy` adaptively updates posterior mass on personas after each answer, so when the model is well specified, it can rapidly concentrate on the right personas and select highly informative questions. This is what we see in the synthetic experiments. But when the persona bank is misspecified relative to real users, the same adaptivity can make `greedy` more sensitive to early noisy answers, steering it toward locally informative but globally suboptimal queries. `nonadaptive` design is less personalized, but also less reactive, and can therefore be more robust.
>
> > **Could you provide runtime comparisons?**
>
> We optimized all methods using standard engineering techniques including contiguous NumPy arrays, Numba JIT, and Joblib parallelization. On WorldValuesBench with 88,459 users and 91 questions, the runtimes in minutes are:
>
> | Method       | Runtime (minutes) |
> | ------------ | ----------------: |
> | random_fixed |              0.48 |
> | nonadaptive  |              0.49 |
> | random       |              0.50 |
> | cat_grm      |             20.78 |
> | cat_gpcm     |             21.98 |
> | greedy       |             42.33 |
> | cat_mgrm     |            100.92 |
> | cat_mgpcm    |            153.61 |
>
> So batch persona-based methods (`random_fixed`, `nonadaptive`, `random`) are extremely fast, one-dimensional CAT (`cat_grm`, `cat_gpcm`) and `greedy` are practical, and multidimensional CAT (`cat_mgrm`, `cat_mgpcm`) is substantially slower. We will report this in the revision.
>
> > **Did you consider latent class analysis (LCA) as a baseline?**
>
> Our model is indeed structurally related to LCA in that both are finite mixtures over categorical responses. However, there are several important differences:
>
> 1. Standard LCA estimates class-conditional response probabilities from **task-specific user data**, whereas our class-conditional response distributions are provided by **LLM-based persona simulations**, making the method usable in cold-start settings and for new questions without calibration data.
> 2. Our model is used not just for static clustering/prediction, but as a **belief state for adaptive querying** with closed-form posterior updates after each answer.
> 3. Because the personas are external, semantically interpretable, and portable across domains, the model can transfer in settings where standard LCA would need to be refit from scratch.
>
> We will acknowledge the structural connection to LCA more explicitly while emphasizing these key differences in the revision.
>
> > **The persona dictionary is fixed; sensitivity is unclear.**
>
> We agree that sensitivity to the persona dictionary is important, and so we conducted three additional sensitivity checks. Please kindly refer to our **response to Reviewer z9a3** for the results. Overall, the current setup remains the strongest. We will incorporate these ablations in the revision.

---

> > ### Author Rebuttal · Reviewer_yZoh · 2026-04-03
> >
> > My concerns have been adequately addressed

---

### Official Review · Reviewer_Mp6Y · 2026-03-12

**Soundness:** 3
**Presentation:** 4
**Significance:** 3
**Originality:** 3
**Overall Recommendation:** 4
**Confidence:** 3

**Summary:**

This paper introduces a framework for adaptive querying under strict question budgets by leveraging LLMs to construct structural Bayesian priors . Instead of querying LLMs in real-time, the authors pre-compute response distributions for a finite dictionary of AI personas across a bank of questions . During live interactions, a new user is modeled as a mixture of these personas, which allows the system to use highly efficient, closed-form Bayesian updates to select the most informative next question. Empirical evaluations on synthetic and real-world datasets demonstrate that this approach outperforms classical Computerized Adaptive Testing (CAT) baselines, particularly in cold-start scenarios . Ultimately, the work successfully bridges the gap between the expressive profiling capabilities of LLMs and the computational tractability of classical statistical experimental design .

**Compliance With Llm Reviewing Policy:**

Affirmed.

**Final Justification:**

The authors have submitted a detailed rebuttal addressing all my questions. Some of the weaknesses remain debatable, but overall I think the merits of this work justify acceptance consideration.

**Key Questions For Authors:**

See the weaknesses above

**Limitations:**

Yes

**Strengths And Weaknesses:**

Strengths:

Soundness: The mathematical formulation is rigorous and elegant, effectively translating the complex outputs of LLMs into tractable, closed-form Bayesian updates .
Presentation: The paper is well-structured and clearly outlines the theoretical framework. The authors also provide insightful and candid ablations, notably observing and explaining how the highly adaptive greedy strategy can overfit on real human data at larger query budgets compared to non-adaptive batch designs .
Significance: The method provides a highly practical solution to the "cold-start" problem inherent in classical Item Response Theory (IRT) and CAT models
Originality:
1. The architectural decision to decouple the LLM from the live interaction loop is a clever synthesis of modern generative AI and classical statistics
2. using readable personas allows stakeholders to view their model’s decision better and represent their space better.

Weaknesses:

Soundness:
1. The primary empirical comparison against classical CAT in a cold-start regime constitutes a somewhat asymmetrical baseline. The proposed model leverages a rich, pre-computed LLM knowledge base, whereas classical CAT structurally requires extensive historical calibration data to function effectively, inevitably disadvantaging the baseline.
2. The method for extracting probability distributions from LLMs relies on direct prompting (e.g., asking for a JSON list of probabilities), which is known to be brittle and sensitive to prompt phrasing. This introduces a risk that the foundational likelihoods of the model may be poorly calibrated.
3. This model leverages the LLM’s pre-existing biases in a constructive way, but can be very dangerous in inducing unwanted biases.
4. This idea sounds like the underlying mechanics of Collaborative Filtering and Lookalike Modeling, the algorithms running behind social networks platforms (matching the user’s own answers to similar “personas” = “other users”), but with synthetic data. I would recommend adding this as a baseline or acknowledging it.

Originality:
1.  While the specific offline-to-online pipeline is neatly executed, the underlying conceptual mechanism closely mirrors standard collaborative filtering and lookalike modeling, simply substituting a database of real historical users with LLM-simulated personas.
2. The paper appropriately acknowledges contemporary works employing LLMs for adaptive querying, yet it lacks direct empirical comparisons against these modern generative baselines, limiting the assessment of its comparative novelty.

---

> ### Author Rebuttal · Authors · 2026-03-30
>
> We thank the reviewer for the careful and encouraging assessment.
>
> > **The CAT comparison is somewhat asymmetrical in a cold-start regime.**
>
> We agree that CAT is not a perfect choice for cold-start regimes, and this is precisely part of the motivation for our work. Our intent in comparing against CAT was not to claim it is ideal for cold-start transfer, but to show that **even when CAT is given substantial calibration data, the persona-based Bayesian prior remains competitive or superior**. In our WorldValuesBench setup, CAT is trained on ~70k users, so the comparison is meaningful as a benchmark against a strong classical adaptive-testing family. We will clarify this framing in the revision.
>
> > **Directly eliciting probabilities from an LLM may be poorly calibrated.**
>
> We agree that this is an important concern. Empirically, the relevant question for our paper is whether the resulting persona-conditioned predictive distributions are useful for adaptive querying. To probe this, we tested alternative constructions, including **deterministic point predictions with injected uniform noise** and **temperature scaling** (please see our **response to Reviewer z9a3**). Both performed worse than the original elicited distributions, suggesting that the direct distribution-elicitation approach is the strongest among the variants we tested. We will make this empirical point clearer in the revision.
>
>
> > **The approach may inherit and amplify unwanted LLM biases.**
>
> We agree that this is an important limitation of any persona-based pipeline built on LLM outputs. We will strengthen the discussion of this issue and identify bias mitigation / calibration of persona priors as an important direction for future work.
>
> > **This seems similar to collaborative filtering (CF) / lookalike modeling.**
>
> We thank the reviewer for raising this. There is a high-level analogy as both relate a user to representative profiles, but the methods are fundamentally different, and we want to make the distinction explicit. Specifically:
>
> 1. CF/lookalike methods are data-driven similarity methods over historical user-item interactions. Our method is a **generative Bayesian model** with explicit likelihoods and posteriors.
>
> 2. CF/lookalike methods, CAT, and standard latent class approaches require observed task-specific data to estimate item parameters or similarities. Our method can handle **new questions with no historical response data**.
>
> 3. Our posterior over personas is a belief state that is updated analytically after each answer, enabling **decision-theoretic query selection**. CF/lookalike methods are usually predictive/retrieval methods, not Bayesian belief-state models for noisy sequential querying.
>
> We will make this distinction clearer, while acknowledging the connection.
>
> > **Why not compare directly to recent LLM-based adaptive-querying methods?**
>
> The main reason is that those methods are **not empirically comparable in our setting**. The works we cited fall into two groups. Some use LLMs primarily as a **language interface** for feature construction or question generation (Handa et al. (2024)), which is a different problem. Others are closer to an **elimination** setting with deterministic likelihoods over a finite hypothesis set (Piriyakulkij et al., 2023; Hu et al., 2024; Maz-zaccara et al., 2024; Kobalczyk et al., 2025). Our setting instead involves **noisy categorical responses**, **posterior uncertainty reduction**, and **closed-form Bayesian design** under misspecification. The elimination-style methods used in those works do not apply to our stochastic setting, and so we discussed them in related work but did not include them as empirical baselines. We will make this non-comparability more explicit.

---

> > ### Author Rebuttal · Reviewer_Mp6Y · 2026-04-02
> >
> > Thank you for the detailed response. I remain leaning positive about this work.

---

### Official Review · Reviewer_z9a3 · 2026-03-12

**Soundness:** 2
**Presentation:** 2
**Significance:** 2
**Originality:** 2
**Overall Recommendation:** 3
**Confidence:** 3

**Summary:**

This paper studies adaptive querying under limited interaction budgets, aiming to infer a user’s responses to unasked questions or related target quantities from only a few queries. The authors propose a Bayesian framework that models user heterogeneity with a finite set of AI personas, where each persona defines a response distribution over questions using offline LLM outputs. By treating the persona index as a discrete latent variable, the method enables closed-form posterior updates and efficient sequential question selection. Experiments on synthetic data and WorldValuesBench show that persona-based querying can outperform traditional CAT baselines, especially at small query budgets.

**Compliance With Llm Reviewing Policy:**

Affirmed.

**Key Questions For Authors:**

Please see the weaknesses.

**Limitations:**

Yes.

**Strengths And Weaknesses:**

S1. The paper introduces a clean formulation that incorporates AI personas into a Bayesian latent-variable model, enabling closed-form posterior updates and making adaptive querying computationally tractable.
S2. By using LLM-generated persona response distributions as structured priors, the method provides a principled way to inject prior knowledge into cold-start settings without requiring model training.
S3. Experiments on both synthetic data and the WorldValuesBench dataset demonstrate that the proposed persona-based querying approach can reduce prediction error compared to traditional CAT baselines, particularly when the number of allowed queries is small.
W1. The core idea mainly reduces to modeling the latent user variable as a finite set of personas, which effectively turns the problem into a discrete latent class model with LLM-generated priors. While the integration with adaptive querying is clean, the overall methodological contribution appears incremental relative to existing latent-variable or mixture-based approaches.
W2. The main experimental results are primarily presented through curves, which makes it difficult to directly compare performance across methods. A clearer presentation using tables (e.g., reporting final metrics at representative budgets) would improve readability. In addition, several details about the experimental setup and implementation—such as how persona priors are constructed and how baselines are calibrated—are not sufficiently clear.
W3. The effectiveness of the approach relies heavily on the quality of the LLM-generated persona dictionary, yet the paper provides limited analysis of how sensitive the method is to the choice, size, or construction of these personas. More ablations or comparisons with alternative prior constructions would strengthen the empirical support.

---

> ### Author Rebuttal · Authors · 2026-03-30
>
> We thank the reviewer for the thoughtful feedback and constructive criticism.
>
> > **Method seems close to a discrete latent class / mixture model; originality appears incremental.**
>
> We believe our main contribution is showing how **LLM-generated personas can be turned into an explicit Bayesian prior that supports closed-form posterior updates and efficient adaptive querying in noisy cold-start settings**. With the recent popularity of AI personas, our paper is meant not only as a modeling observation, but as an **end-to-end operationally simple recipe** that achieves both expressiveness and tractability. We will sharpen this positioning in the revision.
>
> > **The main results are mostly curves; tables at representative budgets would make comparisons easier.**
>
> We will add summary tables at representative budgets to the main text. For illustration, below are **log loss** values at representative budgets on **10,000 randomly sampled real users from WorldValuesBench** (therefore the slight numerical differences from the curves in the paper) with standard errors in parentheses:
>
> | Method      |          Budget 5 |         Budget 15 |         Budget 30 |
> | ----------- | ----------------: | ----------------: | ----------------: |
> | cat_mgrm    |     1.041 (0.007) |     1.027 (0.007) |     1.008 (0.007) |
> | nonadaptive |     1.021 (0.006) | **0.975 (0.005)** | **0.961 (0.005)** |
> | greedy      | **1.003 (0.005)** |     0.987 (0.006) |     0.984 (0.006) |
>
>
> > **Experimental setup and implementation details are not sufficiently clear.**
>
> We thank the reviewer for raising this. The current draft **defers much of the implementation details to the appendix**. Specifically:
>
> * **Appendix A** describes how we obtain response distributions from LLMs;
> * **Bottom of page 6** and **Appendix B** describe the persona dictionary and show the prompts;
> * **Appendix C** gives the details of the ablation studies;
> * **Appendix D** provides a self-contained overview of CAT and details our implementation.
>
> To answer the reviewer's questions more directly: the persona dictionary is built from Twin2K500 personas, and for each persona-question pair we prompt GPT-5-mini to produce a response distribution. For CAT, because robust open-source Python package for multidimensional polytomous CAT is limited, we implemented GRM, GPCM, MGRM, and MGPCM ourselves using marginal maximum likelihood with grid-based posterior inference. We will make these details easier to locate in the revision.
>
> > **W3: The method may be sensitive to persona choice / size / construction; more ablations would help.**
>
> We agree that sensitivity to the persona dictionary is important. In response, we conducted three additional checks:
> 1. **compressing the persona dictionary by clustering**,
> 2. replacing elicited distributions with **deterministic answers plus injected uniform noise**, and
> 3. **temperature scaling** of the LLM-generated distributions.
>
> The results for `nonadaptive` and `greedy` are presented below (all numbers are **log loss on 10,000 randomly sampled users from WorldValuesBench**). Overall, the current setup remains the strongest. While moderate compression can reduce dictionary size with limited loss, deterministic with $\epsilon$-noise or temperature perturbations generally degrade performance. We will incorporate these ablations in the revision.
>
> **Nonadaptive:**
>
> | Variant       |          Budget 5 |         Budget 15 |         Budget 30 |
> | ------------- | ----------------: | ----------------: | ----------------: |
> | current setup | **1.021 (0.006)** | **0.975 (0.005)** | **0.961 (0.005)** |
> | cluster = 50  |     1.022 (0.006) |     0.986 (0.005) |     0.970 (0.005) |
> | cluster = 200 |  1.022 (0.006) |     0.977 (0.005) |     0.962 (0.005) |
> | deterministic w/ $\epsilon$=0.1 |     1.028 (0.010) |     1.271 (0.013) |     1.351 (0.014) |
> | deterministic w/ $\epsilon$=0.3 |     1.080 (0.006) |     1.080 (0.007) |     1.115 (0.008) |
> | temp = 0.5 |     **1.004 (0.008)** |     1.021 (0.009) |     1.034 (0.010) |
> | temp = 2   |     1.096 (0.003) |     1.075 (0.003) |     1.070 (0.003) |
>
> **Greedy:**
>
> | Variant       |          Budget 5 |         Budget 15 |         Budget 30 |
> | ------------- | ----------------: | ----------------: | ----------------: |
> | current setup | **1.003 (0.005)** |     0.987 (0.006) | **0.984 (0.006)** |
> | cluster = 50  |     1.011 (0.005) |     0.995 (0.006) |     0.984 (0.006) |
> | cluster = 200 |  **1.003 (0.005)** | **0.985 (0.006)** |     0.985 (0.006) |
> | deterministic w/ $\epsilon$=0.1 |     1.217 (0.011) |     1.367 (0.014) |     1.427 (0.014) |
> | deterministic w/ $\epsilon$=0.3 |     1.065 (0.006) |     1.105 (0.007) |     1.137 (0.008) |
> | temp = 0.5 |     1.051 (0.010) |     1.096 (0.011) |     1.107 (0.011) |
> | temp = 2   |     1.098 (0.003) |     1.081 (0.003) |     1.074 (0.003) |

---

> > ### Author Rebuttal · Reviewer_z9a3 · 2026-04-03
> >
> > Most of my concerns are resolved. I will think about re-evaluating this manuscript.

---

### Decision · Program_Chairs · 2026-04-30

**Decision:**

Accept (regular)

**Comment:**

The paper presents a Bayesian adaptive querying framework that uses LLM-generated persona distributions as a structured prior. Users are modeled as mixtures over a finite set of personas, enabling closed-form posterior updates and efficient sequential query selection. This makes Bayesian experimental design practical in cold-start settings.

Reviewers find the work technically sound and well motivated. A key strength is the combination of expressive LLM-based priors with tractable inference. Experiments on synthetic data and WorldValuesBench show consistent gains over CAT baselines, especially at small budgets.

The main concern is novelty. The model is closely related to finite mixture and latent class approaches, with connections to collaborative filtering. However, the contribution lies in casting LLM personas as an explicit Bayesian prior that enables efficient adaptive querying under noise and cold-start conditions.

Some limitations remain, including reliance on LLM-based likelihoods and simplifying independence assumptions, but these do not undermine the empirical results.

After the rebuttal, most concerns were resolved and reviewers are positive overall. I therefore recommend acceptance.